# Robust Heterogeneous Graph Neural Network Explainer with Graph Information Bottleneck

## Abstract

Explaining the prediction process of Graph Neural Network (GNN) is crucial for enhancing network transparency. However, real-world networks are predominantly heterogeneous and often beset with noise. The presence of intricate relationships in heterogeneous graphs necessitates a consideration of semantics during the explanation process, while mitigating the impact of noise remains unexplored. For GNN explainers heavily reliant on graph structure and raw features, erroneous predictions may lead to misguided explanations under the influence of noise. To address these challenges, we propose a Robust Heterogeneous Graph Neural Network Explainer with Graph Information Bottleneck, named RHGIB. We theoretically analyze the power of different heterogeneous GNN architectures on the propagation of noise information and exploit denoising variational inference. Specifically, we infer the latent distributions of both graph structure and features to alleviate the influence of noise. Subsequently, we incorporate heterogeneous edge types into the generation process of explanatory subgraph and utilize Graph Information Bottleneck framework for optimization, allowing the Explainer to learn heterogeneous semantics while enhancing robustness. Extensive experiments on multiple real-world heterogeneous graph datasets demonstrate the superior performance of RHGIB compared to state-of-the-art baselines.

## 1 Introduction

Graph Neural Network (GNN), as a powerful tool for learning from graph-structured data, finds applications in various real-life scenarios, such as social networks (Zhang et al., 2023b), citation networks (Li et al., 2022), and recommendation systems (Gao et al., 2022). GNN integrates node features and graph structural information into message passing algorithms, achieving remarkable performance in numerous tasks like graph classification (Liu et al., 2024), node classification (Luan et al., 2024), and link prediction (Lu et al., 2023). Despite their advantages, the decision-making process of GNN is opaque, lacking interpretable explanations for human understanding (Müller et al., 2024). This opacity hampers their application in critical domains related to fairness, privacy, and security (Wang et al., 2024). Hence, researching the explainability of GNN enables a better understanding of their functioning and facilitates improvements toward beneficial outcomes.

GNN Explainers take the original graph and model as inputs, aiming to identify the critical subgraph that significantly influences predictions. Existing GNN Explainers can be categorized into post-hoc and built-in methods (Yuan et al., 2022; Zhang et al., 2024). Post-hoc methods (Vu & Thai, 2023; Pereira et al., 2023; Huang et al., 2022) apply explanation techniques or build explanation models on the base of well-trained models to measure the contributions of different components, thereby explaining the working mechanisms or decision rationales. Built-in methods (Seo et al., 2024; Zhang et al., 2022b; Yuan et al., 2020) generate explanations during the model training process, where the generated graphs serve as explanations for the target predictions and are expected to satisfy specific task objectives. Since built-in methods are tailored for specific models and require separate training for different scenarios, they lack generalizability, whereas post-hoc explanation methods can be applied to most scenarios. Therefore, in this paper, we focus on studying post-hoc methods. Specifically, we aim to develop a superior GNN Explainer that can generate high-quality explanations across different real-world scenarios.

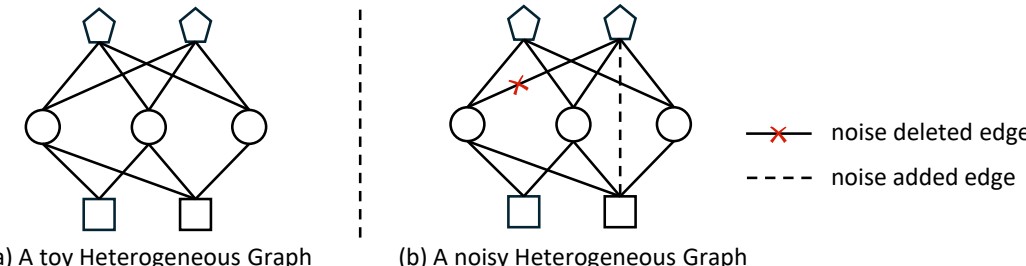

Figure 1: Illustration of a toy heterogeneous graph and the heterogeneous graph with structural noise, where in (a) edges connecting to different types of nodes represent various complex relations on the heterogeneous graph, and in (b) the dashed lines indicate spurious relations added under the influence of noise, and the lines with crosses represent relations disrupted by noise.

Although current GNN explainers have shown excellent performance in certain graph explanation tasks, challenges in real-world still limit their applications. First, real-world graphs are typically heterogeneous, containing multiple types of nodes and edges (Wang et al., 2022; Yang et al., 2020), such as the three node types and two relation types in Figure 1(a), naturally implying their structural complexity. Second, real-world graph data is noisy (Fox & Rajamanickam, 2019; Dai et al., 2022). Specifically, the graph structure may contain edges added or removed due to noise, as illustrated in Figure 1(b). Simultaneously, node features can also be distorted by noise, rendering them unrealistic. Noise poses a critical issue for heterogeneous graphs because the inherent heterogeneity across node types causes noisy edges to carry erroneous heterogeneous relation information, which is further propagated during the message passing process.

Due to the presence of noise, which adds extraneous and unrelated information to the data, the statistical characteristics and distribution of the data are disrupted. This prevents the explainer from effectively learning and extracting the critical patterns in the data, significantly reducing its performance and increasing the risk of generating erroneous explanations. However, no explanation method has investigated noise in heterogeneous graphs. We theoretically demonstrate that the impact of noisy information is amplified by partially heterogeneous graph neural network methods (e.g., meta-paths), thereby interfering with the model decision-making process. Simultaneously, noise intensifies the irregularity of graph structures and alters node importance, rendering conventional explainable methods reliant on strict structural constraints (e.g., size, budget, connectivity) inapplicable (Luo et al., 2020). These methods tend to include noisy edges and exclude correct edges due to the skewed perception of node importance. Thus, adaptively exploring critical subgraphs while managing the irregularities introduced by noisy scenarios presents a significant challenge for explainability in graph neural network.

To address the aforementioned challenges, this paper proposes a Robust Heterogeneous Graph Neural Network Explainer with Graph Information Bottleneck, called RHGIB. We first theoretically demonstrate the amplifying effect of noise information in heterogeneous scenarios. We then employ denoising variational inference to capture robust graph information in the latent variable space. By incorporating the Graph Information Bottleneck principle, we transform the GNN explanation problem into an optimization task, effectively handling irregularities induced by structural noise. Additionally, we propose a relation-based explanation generator that fully considers the complex semantics of heterogeneous graphs during the generation of explanatory subgraphs. To validate RHGIB's explanation capability and effectiveness in handling noise, we evaluate our method on multiple datasets and compare its performance against state-of-the-art baselines.

The contributions of this paper are as follows:

- This is the first work studying the impact of noise on heterogeneous graph explainer, proposing RHGIB to mitigate the performance degradation caused by noise.

- We theoretically prove the amplification effect of existing heterogeneous graph methods on noise and incorporate denoising variational inference to alleviate noise-induced information corruption.

- We propose a novel graph explanation generator based on type attention that incorporates heterogeneous relation learning, effectively capturing complex semantics in the process of explanatory subgraph generation.
- Extensive experiments on three real-world datasets demonstrate RHGIB's superiority and enhanced robustness over state-of-the-art GNN explainers.

## 2 RELATED WORK

**GNN Explainability.** Recently, various approaches have been proposed to explain the predictions of GNN, these approaches can be categorized into post-hoc and built-in method. Common post-hoc methods include perturbation-based (Vu & Thai, 2023; Schlichtkrull et al., 2020) and surrogate model-based (Pereira et al., 2023; Huang et al., 2022) approaches. MixupExplainer (Zhang et al., 2023a) extends the existing GIB framework by introducing label-independent subgraphs during the sampling of explanation subgraphs, thereby obtaining explanations while mitigating the distribution shift phenomenon. GNNExplainer (Ying et al., 2019) learns masks for features and edges by optimizing the masks to obtain the optimal explanation. PGExplainer (Luo et al., 2020) employs a parametric neural network approach to learn the importance of each edge and ultimately selects edges with high importance scores to construct the explanatory subgraph. PGM-Explainer (Vu & Thai, 2020) adopts a Bayesian network formulation, naturally expressing the dependencies between nodes in the form of conditional probabilities. Common built-in methods include prototype learning-based (Seo et al., 2024; Zhang et al., 2022b) and graph generation-based (Yuan et al., 2020) approaches. PGIB (Seo et al., 2024) integrates prototypes into the Graph Information Bottleneck framework, allowing it to learn prototypes based on key subgraphs in the input graph, thereby providing a more accurate explanation of the prediction process. GOAt (Lu et al., 2024) generates explanatory subgraphs by decomposing the model's output into a series of scalar products involving node and edge features, and calculating the contribution of each feature to these scalar products, thereby highlighting the edges that are important for the prediction outcome.

## 3 PROBLEM DEFINITION

In this section, we expound upon the concept of heterogeneous graphs and formally establish the definition of the explanation problem on heterogeneous graphs.

### 3.1 HETEROGENEOUS GRAPH

A heterogeneous graph (HG), denoted as $\mathcal{G} = (\mathbf{A}, \mathbf{X}, \mathcal{A}, \mathcal{R}, \Phi)$, encompasses multiple types of nodes $\mathcal{V}$ and edges $\mathcal{E}$, where $\mathbf{A}$ is the corresponding adjacency matrix, $\mathbf{X}$ represents node features, $\mathcal{A}$ denotes the set of node types, $\mathcal{R}$ signifies the set of edge types, and $\Phi$ represents the set of meta-paths. A meta-path $\phi \in \Phi$ is a path of edges connecting various types of nodes from node $v_1$ to node $v_{l+1}$, such as $\mathcal{A}_1 \xrightarrow{\mathcal{R}_1} \mathcal{A}_2 \xrightarrow{\mathcal{R}_2} \ldots \xrightarrow{\mathcal{R}_l} \mathcal{A}_{l+1}$, where $l$ denotes the length of the meta-path. The label set of $\mathcal{G}$ is denoted as $\mathbf{Y}$, comprising $\mathcal{C}$ categories. Meanwhile, a heterogeneous graph has two mapping functions $\psi(v) : \mathcal{V} \to \mathcal{A}$ and $\varphi(e) : \mathcal{E} \to \mathcal{R}$ that correspond to nodes and edges, respectively.

### 3.2 HETEROGENEOUS GRAPH NEURAL NETWORK EXPLAINER

Given a trained GNN model $f$ as the subject of explanation and a heterogeneous graph $\mathcal{G}$, the objective of the GNN explainer is to identify the most influential subgraph $\mathcal{G}_s = (\mathbf{A}_s, \mathbf{X}, \mathcal{A}_s, \mathcal{R}_s)$. Here, $\mathbf{A}_s$ represents the adjacency matrix formed by nodes $\mathcal{V}_s$ and $\mathcal{E}_s$ which significantly contribute to prediction. For the original prediction of GNN model $f$, it can be formalized as follows:

$$\hat{y} = \underset{c \in \mathcal{C}}{\arg\max} \, P_f(c|\mathbf{A}, \mathbf{X}, \mathcal{A}, \mathcal{R}), \tag{1}$$

where $P_f(\cdot)$ represents the prediction function of the GNN model $f$. Current research indicates that graph structural information is crucial for classification tasks (Luo et al., 2020; Zügner et al., 2018). Therefore, our RHGIB focuses on exploring structural noise when generating explanations. The excellent explanation should contain the most critical information to approximate the predicted

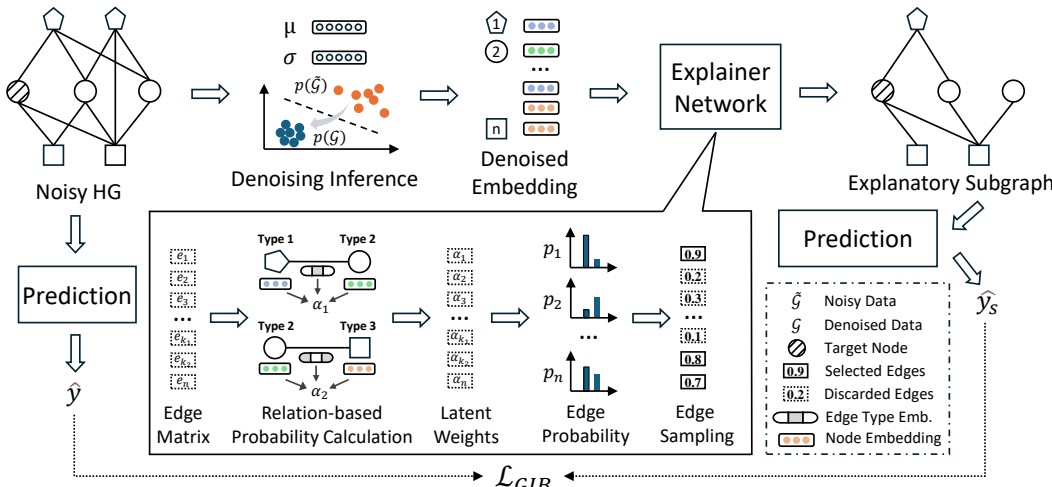

Figure 2: The architecture of our proposed RHGIB. First, the denoised node representations are obtained from the noisy graph via denoising variational inference. Then, the Explainer Network employs the relation-based importance computation method to obtain the weights for different edges. The top k percent of edges are selected as important edges to generate the explanatory subgraph. Finally, the generated explanatory subgraph and the original graph are respectively input into heterogeneous GNN models to obtain predictions, which are used to compute the loss function.

labels and outcome changing when predicting the remaining part of the original graph, which is as follows:

$$\underset{c \in \mathcal{C}}{\arg\max} \, P_f(c|\mathbf{A}_s, \mathbf{X}, \mathcal{A}_s, \mathcal{R}_s) = \hat{y}. \tag{2}$$

## 4 METHODOLOGY

In this section, we formally introduce RHGIB. We first employ the Denoising Variational Inference Graph Encoder to generate a robust representation of the input graph $\mathcal{G}$, and the denoised node embeddings sampled from the latent graph distribution produce edge representations. Subsequently, the Relation-based Explanation Generator incorporates the input edge representations into a heterogeneous relation-based attention learning paradigm to obtain the importance of each edge. The explanation model generates the explanatory subgraph based on the importance scores. Finally, we optimize the proposed method using the Graph Information Bottleneck (GIB) objective. Figure 2 illustrates the framework of RHGIB.

### 4.1 NOISE ANALYSIS AND DENOISING VARIATIONAL INFERENCE

We investigate the impact of noise on different approaches for heterogeneous graph neural network. We categorize common heterogeneous graph neural network into two classes: meta-path-based and neighborhood aggregation-based methods. Meta-path-based methods typically require defining a meta-path $\phi$, and then capturing information along different relations following the meta-path structure, aggregating this information, such as Paths2Pair (Hang et al., 2024) and MAGNET (Wen et al., 2023). Neighborhood aggregation-based methods simultaneously consider the neighbor node types and edge types for each node and use specific aggregation functions to combine information from different types. Common neighborhood aggregation methods include MHGCN (Yu et al., 2022) and Simple-HGN (Lv et al., 2021). However, these two categories of methods differ in their efficiency of noise propagation (Zhang et al., 2022a), and we find that meta-path-based message passing methods amplify the impact of noise.

**Lemma 1 (Perturbation Enlargement Effect.)** *Given a node $v$ from a heterogeneous graph $\mathcal{G}$, when the edges adjacent to $v$ are perturbed, meta-path-based methods can enlarge the disruptive effect of the perturbation.*

**Theorem 1 (Noise Amplification Effect in HG.)** *In HG, compared to neighborhood aggregation-based methods, meta-path-based methods can significantly amplify the effect of noisy edges. Specifically, for a node $v_i$ and a newly added noisy edge $e_{ij}$, the factor by which its influence changes is $\frac{d_{v_i}+k}{d_{v_i}+1}$, where $k$ is the degree of the new neighbor $v_j$ under the noise and $d_{v_i}$ is the degree of $v_i$. When $k > d_{v_i}$, this factor is significantly greater than 1.*

The complete proof of Theorem 1 is provided in Appendix B.

Based on Theorem 1, we employ a neighborhood aggregation method to encode heterogeneous graph and mitigate noise. Given noisy graph data $\tilde{\mathcal{G}}$, our objective is to obtain a denoised version of the standard graph data $\mathcal{G}$. The Variational Graph Auto-Encoder (VGAE) (Kipf & Welling, 2016b) uses variational inference to derive statistical properties of the graph. In VGAE, the statistical data of latent variables can be efficiently inferred from the latent space rather than the observation space, which provides robust graph information. For the standard graph $\mathcal{G}$, VGAE initially generates latent variables $\mathbf{Z}$ from a prior distribution $p(\mathbf{Z})$, such as a Gaussian distribution $\mathcal{N}(\boldsymbol{\mu}, \boldsymbol{\sigma}^2)$. Second, the data $\mathcal{G}$ is generated using a conditional distribution $p(\mathcal{G}|\mathbf{Z})$. VGAE optimizes its parameters by maximizing the likelihood of the observed data, which as follows:

$$\mathrm{KL}(q_\Psi(\mathbf{Z}|\mathcal{G})||p_\theta(\mathbf{Z}|\mathcal{G})) + \mathcal{L}(\Psi, \theta; \mathcal{G}), \tag{3}$$

where $\Psi$ is the encoder and $\theta$ represents the parameters to be optimized. Then, the evidence lower bound $\mathcal{L}(\Psi, \theta; \mathcal{G})$ can be expressed as follows:

$$\mathcal{L}(\Psi, \theta; \mathcal{G}) = \mathbb{E}_{q_\Psi(\mathbf{Z}|\mathcal{G})}[\log \frac{p_\theta(\mathbf{Z}, \mathcal{G})}{q_\Psi(\mathbf{Z}|\mathcal{G})}] = \mathbb{E}_{q_\Psi(\mathbf{Z}|\mathcal{G})}[\log p_\theta(\mathbf{Z}|\mathcal{G})] - \mathrm{KL}(q_\Psi(\mathbf{Z}|\mathcal{G})||p(\mathbf{Z})).$$

Variational inference enhances the model's robustness and generalization capabilities (Fan et al., 2021; Im Im et al., 2017). However, due to the differing distributions between noisy graph data and standard graph data, the obtained distribution tends to align with the noisy distribution, potentially misleading the GNN explainer into generating incorrect explanatory subgraphs. Therefore, we introduce a denoising module during the process of variational inference. The original encoder part is modified to:

$$q'_\Psi(\mathbf{Z}|\mathcal{G}) = \int q_\Psi(\mathcal{G}|\tilde{\mathcal{G}})q(\tilde{\mathcal{G}}|\mathcal{G})\mathrm{d}\tilde{\mathcal{G}}, \tag{4}$$

where $\Psi$ is the encoder based on $\tilde{\mathcal{G}}$. During this process, the evidence lower bound is expressed as:

$$\mathcal{L}_d = \mathbb{E}_{q'_\Psi(\mathbf{z}|\mathcal{G})}[\log \frac{p_\theta(\mathbf{Z}, \mathcal{G})}{q'_\Psi(\mathbf{Z}|\mathcal{G})}]. \tag{5}$$

As we need to derive the distribution of the noisy graph data $\tilde{\mathcal{G}}$, this lower bound can be further refined as:

$$\mathcal{L}_d = \mathbb{E}_{q'_\Psi(\mathbf{z}|\mathcal{G})}[\log \frac{p_\theta(\mathbf{Z}, \mathcal{G})}{q'_\Psi(\mathbf{Z}|\mathcal{G})}] \geq \mathbb{E}_{q'_\Psi(\mathbf{z}|\mathcal{G})}\left[\log \frac{p_\theta(\mathcal{G}, \mathbf{Z})}{q_\Psi(\mathbf{Z}|\tilde{\mathcal{G}})}\right]$$

$$= \mathbb{E}_{q'_\Psi(\mathbf{z}|\mathcal{G})}[\log p_\theta(\mathcal{G}|\mathbf{Z})] - \mathbb{E}_{q(\tilde{\mathcal{G}}|\mathcal{G})}[\mathrm{KL}(q_\Psi(\mathbf{Z}|\tilde{\mathcal{G}}))||p(\mathbf{Z})]. \tag{6}$$

The detailed derivation is in the Appendix C. Compared to VGAE, the denoising variational inference models the posterior probability $p(\mathbf{Z}|\mathcal{G})$ using a Gaussian Mixture Model, whereas VGAE models $p(\mathbf{Z}|\mathcal{G})$ using a Gaussian distribution. Additionally, during the optimization process, there is a constraint that forces $q_\Psi(\mathbf{Z}|\tilde{\mathcal{G}})$ to approximate the standard Gaussian distribution $p(\mathbf{Z})$. Consequently, our method can significantly improve the model's robustness and produce high-quality graph data. We further employ the Monte Carlo sampling method to approximate the objective, which can be effectively optimized using gradient descent as follows:

$$\mathcal{L}_d \approx \frac{1}{K} \sum_{k=1}^{K} \log \frac{p_\theta(\mathcal{G}, \mathbf{Z})}{q_\Psi(\mathbf{Z}|\tilde{\mathcal{G}})}, \tag{7}$$

where $K$ is the number of samples sampled during the simulation.

After denoising variational inference, we input the sampled robust representations $\mathbf{Z}$ into the Relation-based Explanation Generator, where the complex semantics on the heterogeneous graph are learned. Before delving into that, we introduce the Graph Information Bottleneck.

## 4.2 GRAPH INFORMATION BOTTLENECK

As mentioned in the introduction, noise exacerbates the irregularity of graph structures and alters node importance. Therefore, previous methods imposing structural regularity constraints on explanatory subgraphs are infeasible under noise influence. We exploit the Graph Information Bottleneck (GIB) to enable the explainer network to adaptively handle structural irregularities. The objective of GIB is to obtain the optimal explanatory subgraph $\mathcal{G}_s$. From an information-theoretic perspective, GIB limits the amount of information carried by the explanatory subgraph $\mathcal{G}_s$, rather than imposing simple structural constraints. Simultaneously, nodes may require scattered edges across the graph to jointly explain their predictive function, rather than constraining connectedness. For instance, in molecular graphs, a molecule may contain multiple functional groups scattered throughout the graph, collectively determining its properties. Consequently, GIB adaptively explores the entire graph without imposing any potentially biased restrictions. GIB can be formulated as:

$$\min_{\mathcal{G}_s \subset G} -\mathrm{I}(\hat{y}; \mathcal{G}_s) + \beta\, \mathrm{I}(\mathcal{G}; \mathcal{G}_s), \tag{8}$$

where $\mathrm{I}(\cdot; \cdot)$ denotes mutual information, and $\beta$ is the Lagrangian multiplier controlling the trade-off between the two terms. Since the information in $\mathcal{G}_s$ can be continually optimized, the explain task can be characterized as an optimization task guided by GIB.

The GIB principle aims to obtain the minimum sufficient information about the graph $\mathcal{G}$. The first term maximizes the mutual information between the label and the explanatory subgraph, ensuring $\mathcal{G}_s$ contains as much information about the label as possible. The second term minimizes the mutual information between the input graph and the explanatory subgraph, ensuring $\mathcal{G}_s$ contains the minimum information about the input graph. Next, we introduce the Relation-based Explanation Generator, describing how each term is optimized during training under the GIB principle.

## 4.3 RELATION-BASED EXPLANATION GENERATOR

We first assume the explanatory subgraph is a Gilbert random graph (Gilbert, 1959), where edges are conditionally independent. Following the literature (Luo et al., 2020), we define an adjacency matrix-like edge matrix $E_s$, where each element $e_{ij}$ is a binary variable indicating whether the edge is included in the subgraph. When there is an edge $(i, j)$ from $v_i$ to $v_j$, $e_{ij} = 1$, otherwise $e_{ij} = 0$. Based on this, the random graph variable can be factorized as:

$$p(\mathcal{G}) = \prod_{(i,j) \in E_s} p(e_{ij}), \tag{9}$$

where $e_{ij}$ is a binary variable following a Bernoulli distribution $\mathrm{Bern}(\pi_{ij})$, and $p(e_{ij})$ denotes the probability of the edge $(i, j)$ existing. Since $e_{ij}$ is discrete, we apply a reparameterization trick to relax it into a continuous variable between 0 and 1 for ease of optimization, as follows:

$$e_{ij} = \mathrm{Sigmoid}\left(\frac{\log \epsilon - \log(1 - \epsilon) + \alpha_{ij}}{\tau}\right), \epsilon \sim \mathrm{Uniform}(0, 1), \tag{10}$$

where $\tau$ is a temperature coefficient to smooth the optimization, and $\alpha_{ij}$ is a heterogeneous attention coefficient. When let $\alpha_{ij} = \log\frac{\pi_{ij}}{1 - \pi_{ij}}$, we have $\lim_{\tau \to 0} p(e_{ij} = 1) = \frac{\exp(\alpha_{ij})}{1 + \exp(\alpha_{ij})}$, so we can obtain the explanatory subgraph $\mathcal{G}_s$ since $p(e_{ij} = 1) = \pi_{ij}$.

To capture the rich semantics in heterogeneous graphs, merely considering pairwise relationships between nodes is insufficient. Thus, we incorporate heterogeneous semantics learning into the explanatory subgraph generation process. Inspired by (Lv et al., 2021), we extend the standard graph attention mechanism by incorporating edge type information into the attention computation. Specifically, we assign an edge type embedding $\mathbf{r}_{\varphi(e)}$ for each edge type $\varphi(e)$, and simultaneously utilize the edge type embeddings and node embeddings to compute $\alpha_{ij}$:

$$\alpha_{ij} = \frac{\exp\left(\mathrm{ReLU}\left(\boldsymbol{a}^T[\boldsymbol{W}\boldsymbol{z}_i \| \boldsymbol{W}\boldsymbol{z}_j \| \boldsymbol{W}_r \boldsymbol{r}_{\varphi(e_{ij})}]\right)\right)}{\sum_{k \in \mathcal{N}_i} \exp\left(\mathrm{ReLU}\left(\boldsymbol{a}^T[\boldsymbol{W}\boldsymbol{z}_i \| \boldsymbol{W}\boldsymbol{z}_k \| \boldsymbol{W}_r \boldsymbol{r}_{\varphi(e_{ik})}]\right)\right)}, \tag{11}$$

where $\boldsymbol{W}_r$ is a learnable weight matrix for type embeddings. Edge type embedding is a one-hot encoding derived from each edge type. Based on Eq. 11, we obtain a probability matrix $\mathbf{M_p}$. The

$(i, j) - th$ element of $\mathbf{M_p}$ represents the probability of the existence of $e_{ij}$. In order to generate the explanation subgraph $\mathcal{G}_s$, we can sample from $\mathbf{M_p}$, as shown below:

$$\mathcal{G}_s = (\mathbf{A_s} = \mathbf{M_p} \odot \mathbf{A}, \mathbf{X}, \mathcal{A}_s, \mathcal{R}_s). \tag{12}$$

However, as is well known, mutual information is challenging to compute, especially in continuous and high-dimensional spaces. We derive an upper bound for GIB through the extension of Jensen's inequality and marginal distributions. Eq. 8 can be written as:

$$-\mathrm{I}(\hat{y}; \mathcal{G}_s) + \beta \, \mathrm{I}(\mathcal{G}; \mathcal{G}_s) \leq -\mathbb{E}_{p(\mathcal{G}_s, \hat{y})}\big[\log p_f(\hat{y}|\mathcal{G}_s)\big] + \mathrm{H}(\hat{y}) + \beta\mathbb{E}_{p(\mathcal{G})}\big[\mathrm{KL}(p_\alpha(\mathcal{G}_s|\mathcal{G})||q(\mathcal{G}_s))\big],$$

where $f$ is the GNN model and $\alpha$ is the explain model, see Appendix C for detailed derivation. Since $\mathrm{H}(\hat{y})$ is constant, the objective function can be expressed as follows:

$$\mathcal{L}_{GIB} = -\mathbb{E}_{p(\mathcal{G}_s, \hat{y})}\big[\log p_f(\hat{y}|\mathcal{G}_s)\big] + \beta\mathbb{E}_{p(\mathcal{G})}\big[\mathrm{KL}(p_\alpha(\mathcal{G}_s|\mathcal{G})||q(\mathcal{G}_s))\big].$$

Finally, RHGIB jointly optimizes the objectives of VGAE and GIB, and the overall objective function is as follows:

$$\mathcal{L} = \mathcal{L}_d + \mathcal{L}_{GIB}. \tag{13}$$

### 4.4 COMPLEXITY ANALYSIS.

The cost of each iteration comprises two parts: (1) the variational inference process and (2) the explanation generation. The time complexity of the first step is $O(N^2 + E)$, and the space complexity is $O(N)$, as this step requires storing the new high-quality node representations. The time complexity of the second step is $O(E)$, and the space complexity is $O(E)$. Therefore, the overall time complexity of RHGIB is $O(N^2 + E)$, and the space complexity is $O(N + E)$.

## 5 EXPERIMENT

In this section, we evaluate the performance of the proposed RHGIB and state-of-the-art baselines on the node classification task. We then analyze the contributions of different components of RHGIB and demonstrate why RHGIB is robust to noise and capable of generating explanations that incorporate heterogeneous information.

Table 1: The comparison of RHGIB and baselines under different ratios of random structural noise. We use bold font to mark the best score. The second best score is marked with underline.

| Dataset | Noise Ratio | 10% | | 20% | | 30% | | 40% | |
|---|---|---|---|---|---|---|---|---|---|
| | | MAE | RMSE | MAE | RMSE | MAE | RMSE | MAE | RMSE |
| DBLP | PGExplainer | 1.2158 | 1.6775 | 1.2179 | 1.6815 | 1.2449 | 1.6999 | 1.2451 | 1.7060 |
| | GNNExplainer | 0.8530 | 1.2968 | 0.9080 | 1.3072 | 1.2613 | 1.8470 | 1.3388 | 1.9043 |
| | PGM-Explainer | 1.0704 | 1.3855 | 1.2046 | 1.5280 | 1.3313 | 1.6497 | 1.3401 | 1.6681 |
| | V-InfoR | 1.1930 | 1.6481 | 1.1960 | 1.6511 | 1.2312 | 1.6781 | 1.2530 | 1.6885 |
| | PGE-Relation | 0.8719 | 1.2814 | 0.8896 | 1.2859 | 1.1913 | 1.6532 | 0.9268 | 1.3100 |
| | RHGIB | **0.8359** | **1.2416** | **0.8743** | **1.2750** | **0.8827** | **1.2792** | **0.9014** | **1.2889** |
| ACM | PGExplainer | 0.7624 | 1.0258 | 0.7751 | 1.0307 | 0.7867 | 1.0423 | 0.7913 | 1.0431 |
| | GNNExplainer | 0.3449 | 0.6831 | 0.3951 | 0.7791 | 0.5087 | 0.9506 | 0.6496 | 1.1306 |
| | PGM-Explainer | 0.2155 | **0.5121** | 0.3732 | 0.6893 | 0.5932 | 0.8793 | 0.7932 | 1.0766 |
| | V-InfoR | 0.7639 | 1.0145 | 0.7913 | 1.0366 | 0.8064 | 1.0705 | 0.8154 | 1.0786 |
| | PGE-Relation | 0.8091 | 1.0740 | 0.8183 | 1.0778 | 0.8220 | 1.0731 | 0.8310 | 1.0856 |
| | RHGIB | **0.2129** | 0.5662 | **0.2483** | **0.6177** | **0.3140** | **0.6669** | **0.3163** | **0.6909** |
| Freebase | PGExplainer | 0.7189 | 1.0616 | 0.7237 | 1.0635 | 0.7285 | 1.0689 | 0.7370 | 1.0803 |
| | GNNExplainer | 0.9012 | 1.2886 | 0.9108 | 1.2983 | 0.9126 | 1.2995 | 0.9378 | 1.3217 |
| | PGM-Explainer | 0.9190 | 1.2432 | 0.9401 | 1.2549 | 0.9530 | 1.2747 | 0.9587 | 1.2838 |
| | V-InfoR | 0.5957 | 1.0375 | 0.6822 | 1.1172 | 0.7249 | 1.1487 | 0.7894 | 1.1929 |
| | PGE-Relation | 0.7760 | 1.1064 | 0.7812 | 1.1117 | 0.7908 | 1.1200 | 0.8030 | 1.1315 |
| | RHGIB | **0.3885** | **0.8251** | **0.4441** | **0.8854** | **0.4694** | **0.9035** | **0.4880** | **0.9217** |

## 5.1 EXPERIMENT SETTINGS

**Datasets and Baselines.** We evaluate the effectiveness of our RHGIB on three real-world datasets, including two academic citation datasets (DBLP and ACM) and a knowledge graph dataset (Freebase). Since there are no existing robust heterogeneous explainer, we select three types of baselines: the surrogate method PGM-Explainer, the perturbation-based methods GNNExplainer and PGExplainer, and the V-infor method studying robustness on homogeneous graphs. Additionally, we extend our Relation-based importance computation module to PGExplainer, denoted as PGE-Relation, for comparison.

**Evaluation.** The evaluation of explainer performance is based on the generated explanatory subgraphs, assessing their contribution to the original prediction. We adopt two metrics: fidelity and sparsity. Fidelity measures the decrease in prediction confidence after removing the explanation from the input graph, while sparsity measures the ratio of remaining edges in the explanatory subgraph $\mathcal{G}_s$ relative to the input graph. In our experiments, we use the Mean Absolute Error (MAE, $\frac{1}{N} \sum_{i=1}^{N} \left| \mathbb{I}(\hat{y}_i = y_i) - \mathbb{I}(\hat{y}_i^{\mathcal{G}_s} = y_i) \right|$) and Root Mean Squared Error (RMSE, $\sqrt{\frac{1}{N} \sum_{i=1}^{N} (\mathbb{I}(\hat{y}_i = y_i) - \mathbb{I}(\hat{y}_i^{\mathcal{G}_s} = y_i))^2}$) as proxy measures for fidelity, and compare the performance of different baselines across varying sparsity levels, where $N$ is the number of nodes or graphs, $\hat{y}_i$ is the original prediction result, and $\hat{y}_i^{\mathcal{G}_s}$ is the prediction result obtained by the explanatory subgraph.

**Implementation Details.** We conduct experiments under different proportions of random noise scenarios. Noise is added to both the training set and the test set to restore the real scene. For the baselines, we select the best-performing parameters for heterogeneous datasets based on the original settings. We chose the most basic HGNN architecture which only contains GCN (Kipf & Welling, 2016a) and relational learning modules as the base model. Each experiment is repeated 5 times, and we report the mean and variance as the results. Descriptions of the variance, datasets, baselines, base heterogeneous GNN model, and parameter settings are provided in the Appendix D.

## 5.2 OVERALL PERFORMANCE UNDER RANDOM NOISE

Table 1 shows the experimental results on the heterogeneous graphs with different ratios of random structural noise. We randomly select and flip edges based on the noise ratio, thereby augmenting or disrupting relations in the heterogeneous graph. We find that RHGIB outperforms other baselines in most experimental results, achieving the best performance on the DBLP and Freebase datasets. Taking 30% noise ratio as an example, RHGIB shows 25.9% lower MAE and 22.4% lower RMSE than the second-best method on the DBLP dataset, 38.2% lower MAE and 24.1% lower RMSE on the ACM dataset, and 35.2% lower MAE and 15.4% lower RMSE on the Freebase dataset. We can observe that PGE-Relation achieves second best performance multiple times on the DBLP dataset and outperforms many baselines on other datasets, demonstrating the effectiveness of our proposed heterogeneous semantic learning module in considering rich semantics on heterogeneous graphs. Due to the similar edge type distribution in the DBLP dataset, the dataset exhibits higher heterogeneity, which enhances the module's ability to capture heterogeneous information. Simultaneously, as a plug-and-play module, it can be conveniently extended to other parameterized explanation methods for generating explanations on heterogeneous graphs. On the medium and small-scale datasets DBLP and ACM, explanation methods based on raw features (e.g., GNNExplainer) are more susceptible to noise, potentially because raw features are more easily affected in smaller graphs. Since RHGIB generates latent variational graph representations, it can better mitigate the influence of noise, which is also why the latent representation-based explainer V-InfoR performs well in multiple scenarios. Under the guidance of Graph Information Bottleneck, our method can adaptively select important edges while excluding redundant and noisy edges, thereby generating the best explanations for the prediction model.

## 5.3 FIDELITY-SPARSITY ANALYSIS

Next, we further investigate RHGIB's performance at different sparsity levels. We provide the Fidelity-Sparsity curve on the DBLP dataset as shown in Figure 3. It can be observed that RHGIB

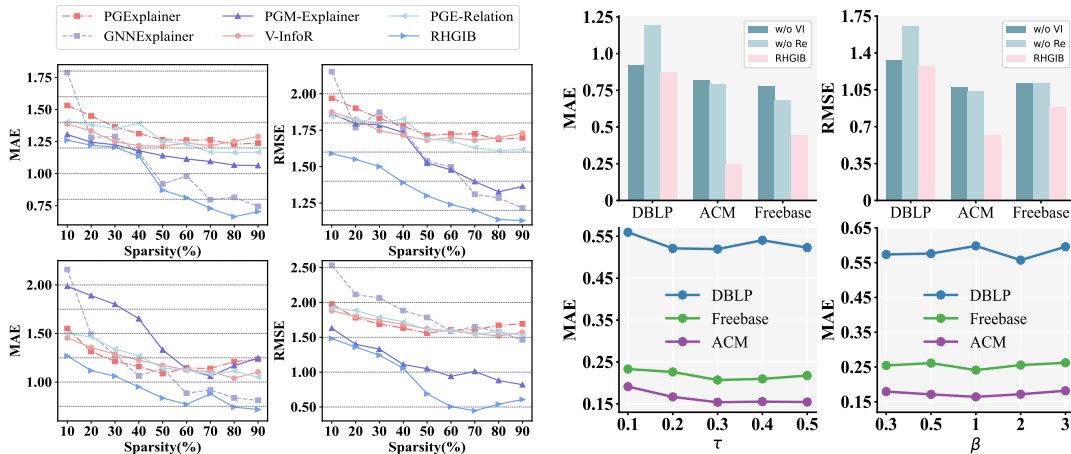

Figure 3: Fidelity-Sparsity Curve on the DBLP dataset. The first row is the result without noise, and the second row is the result with 20% random noise added.

Figure 4: Ablation study on three datasets and the influence of hyperparameters $\tau$ and $\beta$ on RHGIB.

consistently outperforms other baselines across all sparsity levels, indicating that our method can generate the best explanations. As the sparsity increases from 0 to 1, the overall trend of all curves is downward, i.e., decreasing error. When the sparsity is extremely low, e.g., 10%, our method significantly outperforms other baselines, suggesting that RHGIB can identify the truly critical subgraphs. We further find that although the overall performance improves as the sparsity level increases, there are still some cases where the performance drops with increasing sparsity, such as PGExplainer. We conjecture that this may be because in the subgraph generation process, when the sparsity increases to a point where all edges with high importance scores have been selected, forcing higher sparsity will begin to select unimportant edges, which can be viewed as noisy edges, leading to degraded performance. As the sparsity continues to increase, this adverse effect is offset.

## 5.4 Ablation Study

In this section, we investigate the contributions of different components in RHGIB. Specifically, we study (a) the effectiveness of the denoising variational inference module, and (b) the effectiveness of the relation-based importance module. We use 'w/o VI' to denote the model without the denoising variational inference module, and 'w/o Re' to denote the model without the relation-based importance module. For the latter case, we replace it with the common concatenation operation, i.e., $\alpha_{ij} = \text{MLP}[(\mathbf{z}_i, \mathbf{z}_j)]$. The experiments are conducted under 20% random noise, and the first row of Figure 4 shows the results after ablation. We find that without the denoising variational inference module, the model relies on the original features and graph structure for prediction, failing to mitigate the influence of noise, leading to performance degradation. When the model loses the ability to learn heterogeneous relationships, the process of generating explanation subgraphs struggles to recognize the complex semantics in heterogeneous graphs. All edges are treated as the same type, and the model explains solely based on node interactions. This demonstrates the necessity of our proposed relation importance module.

## 5.5 Hyperparameter Analysis

We further analyze the impact of two parameters $\tau$ and $\beta$ on model performance. $\tau$ controls the approximation degree of $e_{ij}$ distribution to the Bernoulli distribution, ranging within $[0.1, 0.5]$. $\beta$ balances the information recovery strength (i.e., $\min -\text{I}(\hat{y}; \mathcal{G}_s)$) and information filtering strength (i.e., $\min \text{I}(\mathcal{G}; \mathcal{G}_s)$) in the optimization objective, and we select values from $\{0.3, 0.5, 1, 2, 3\}$. The second row of Figure 4 shows the effects of these hyperparameters on RHGIB across three datasets. We can observe that the optimal value of $\tau$ does not vary significantly across datasets, but the best results all appear around 0.3. That is, when $\tau = 0.3$, the continuity and approximation degree

in Eq. 10 reach the best trade-off. Secondly, RHGIB is not very sensitive to $\beta$ that controls the constraint strength in Eq. 13, validating that our used GIB constraint can adapt to different data scenarios and achieve superior performance.

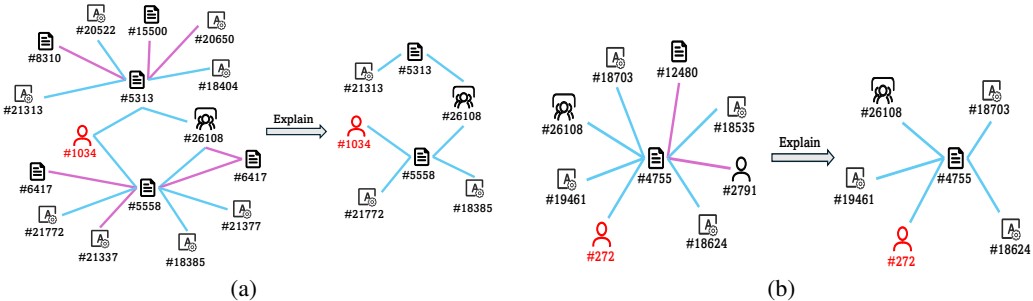

(a)                 (b)

Figure 5: Case study of RHGIB on the DBLP dataset. Purple edges represent noisy edges, blue edges represent normal edges, and red nodes represent the target nodes.

## 5.6 CASE STUDY

We conducted two case studies to visualize the process of RHGIB, with Figure 5 displaying examples from the DBLP dataset. In both examples, normal edges (blue) are explained while retaining only the important edges, and noisy edges (purple) are successfully excluded by the model. As shown in Figure 5(a), when explaining the prediction result for the target node Author-1034, RHGIB successfully identified the important edge <Author-1034, Paper-5558>. This step is noteworthy because Author-1034 is only connected to Paper-5313 and Paper-5558. Due to their weights in message passing, many explainers would mistakenly consider both edges in the explanation subgraph, while RHGIB only recognized the most important edge for the prediction. Additionally, RHGIB successfully excluded all noisy edges. The noisy edges <Paper-5558, Term-21337> and <Paper-5313, Term-20650> share the same relation type with existing edges, making them prone to being mixed with the original heterogeneous semantics. However, they are successfully excluded after the explanation by RHGIB. In Figure 5(b), when explaining the prediction result for the target node Author-272, the explanation subgraph captured all key edges. For the noisy edge <Paper-4755, Paper-12480>, which introduced new heterogeneous semantics that could interfere with the predictions of the base model, this edge was successfully excluded following the RHGIB explanation.

## 6 CONCLUSION

In this work, we focus on the problem of explaining heterogeneous graph neural network under noise. We are the first to study this problem, theoretically proving that heterogeneous graph neural network have an amplifying effect on noise, and propose RHGIB to mitigate the influence of noise and obtain explanatory subgraphs based on heterogeneous relations. Specifically, RHGIB employs denoising variational inference to obtain robust graph representations and parameterizes the explanatory subgraph generation process with neural networks. It integrates rich relation information to capture the complexity of diverse node and edge types. Moreover, RHGIB can explain predictions at the node, edge, and graph levels. Extensive experiments on real-world datasets demonstrate RHGIB's superiority over other state-of-the-art baselines. For future work, we plan to further extend RHGIB to dynamic graphs by incorporating dynamic information into the explanation generation process, further broadening RHGIB's applicability.

## 7 REPRODUCIBILITY

We detail the model design in the paper, including denoising variational inference (Sec. 4.1), the relation-based explanation generator (Sec. 4.3), and the optimization objective (Eq. 13). In Appendix B, we provide a detailed proof of Theorem 1, and in Appendix C, we derive Eq. 4, Eq. 6, and Eq. 13. The experimental setup is explained in Sec. 5.1 and Appendix D. The code is available at: https://anonymous.4open.science/r/RHGIB-EBD0.

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

## A    RELATED WORK ON HETEROGENEOUS GRAPH NEURAL NETWORKS

Heterogeneous Graph Neural Networks can be categorized into meta-path-based methods and neighborhood aggregation-based methods. Meta-path-based methods typically decompose heterogeneous graphs into multiple homogeneous subgraphs using predefined meta-paths, thereby capturing specific types of heterogeneous semantics. Message passing is then performed within each subgraph, and the messages are subsequently aggregated. Common methods in this category include HAN (Wang et al., 2019), MAGNN (Fu et al., 2020), and SeHGNN (Yang et al., 2023). On the other hand, neighborhood aggregation-based methods usually aggregate information directly from neighbors and apply specific aggregation strategies based on node types. Examples of methods in this category include RGCN (Schlichtkrull et al., 2018), NARS (Yu et al., 2020), and Simple-HGN (Lv et al., 2021).

## B    PROOF OF THEOREM 1

In Graph Neural Network, a node representation is typically updated by aggregating information from its neighboring nodes. This process can be described as a message passing mechanism, where each node receives messages from neighboring nodes and updates its representation based on these messages. To avoid cases where the influence is overly amplified during the aggregation process, the messages from neighboring nodes are typically normalized. A common normalization approach is to multiply each neighbor message by the inverse of its degree. Assuming that each node influence on neighbors is equal, a higher-degree node will distribute its influence evenly among all neighbors. Therefore, the influence received by each neighbor should be proportional to the inverse of the node degree. In contrast, in random walk models, the transition probability between nodes is inversely proportional to the node degree. That is, the probability of a node reaching a particular neighbor is the inverse of its degree.

Given a heterogeneous graph $\mathcal{G}$, let $v_i$ be a node with degree $d_{v_i}$. A noisy edge $e_{ij}$ is added to the graph, where $v_j$ is a new neighbor with degree $d_{v_j}$ and $k$ specific-type neighbors that match a given meta-path $\phi$.

For meta-path-based methods:

(a) Before adding the noisy edge, the influence of $v_i$ is assumed to be a combination of the influences from its $d_{v_i}$ existing neighbors $v_1, v_2, ..., v_{d_{p_i}}$ in $\mathcal{G}$. The influence of each neighbor $v_n$ on $v_i$ can be represented as:

$$I_{\text{ori1}} = \sum_{n=1}^{d_{v_i}} \frac{1}{d_{v_i}} \tag{14}$$

(b) After adding the noisy edge $(v_i, v_j)$, $v_i$ is directly connected to $v_j$, and the influence of $v_j$ will propagate to its $k$ neighbors. The influence on each neighbor of $v_j$ changes in the following manner $v_i$:

$$I_{\text{new1}} = \sum_{i=1}^{d_{v_i}} \frac{1}{d_{v_i}+1} + \frac{k}{d_{v_i}+1} = \frac{d_{v_i}}{d_{v_i}+1} + \frac{k}{d_{v_i}+1} = \frac{d_{v_i}+k}{d_{v_i}+1} \tag{15}$$

For neighborhood aggregation-based methods:

(a) Before adding the noisy edge, the influence on the direct neighbors of $v_i$ is given by:

$$I_{\text{ori2}} = \sum_{n=1}^{d_{v_i}} \frac{1}{d_{v_i}} \tag{16}$$

(b) After adding the noisy edge $(v_i, v_j)$, the neighbors of $v_i$ increase to $d_{v1}+1$, and the influence on each of its neighbors changes to:

$$I_{\text{new2}} = \sum_{n=1}^{d_{v_i}+1} \frac{1}{d_{v_i}+1} \tag{17}$$

The multiplicative relationship $\xi$ of the influence propagation between the meta-path-based method and the neighborhood aggregation method is:

$$\xi = \frac{\frac{I_{\text{new1}}}{I_{\text{ori1}}}}{\frac{I_{\text{new2}}}{I_{\text{ori2}}}} = \frac{\frac{d_{v_i}+k}{d_{v_i}+1}}{1} = \frac{d_{v_i}+k}{d_{v_i}+1} \tag{18}$$

Consequently, when $k > d_{v_i}$, the multiplicative factor $\xi$ is significantly greater than 1. This indicates that in general heterogeneous graphs, meta-path-based approaches are far more susceptible to the influence of noisy edges compared to neighborhood aggregation-based approaches. This substantiates that meta-path-based methods can significantly amplify the effect of noisy edges to a greater extent than neighborhood aggregation methods.

## C  DETAILED DERIVATION

First, we give the detailed derivation of Eq. 4. We introduce the Kullback-Leibler (KL) divergence. The KL divergence is a measure used to quantify the difference between two probability distributions. Let us consider two continuous random variables with probability distributions $P$ and $Q$, and their corresponding probability density functions denoted as $p(x)$ and $q(x)$, respectively. If we aim to approximate $p(x)$ using $q(x)$, the KL divergence can be expressed as:

$$\mathrm{KL}(P||Q) = \int p(x) \log \frac{p(x)}{q(x)} dx. \tag{19}$$

Because the logarithmic function is convex, the value of KL divergence is nonnegative. Then, Eq. 4 can be written as:

$$\begin{aligned}
\mathcal{L}(\Psi, \theta; \mathcal{G}) &= \mathbb{E}_{q_\Psi(\mathbf{Z}|\mathcal{G})}[\log \frac{p_\theta(\mathbf{Z}, \mathcal{G})}{q_\Psi(\mathbf{Z}|\mathcal{G})}] \\
&= \mathbb{E}_{q_\Psi(\mathbf{Z}|\mathcal{G})}[\log p_\theta(\mathbf{Z}|\mathcal{G}) \cdot \frac{p(\mathbf{Z})}{q_\Psi(\mathbf{Z}|\mathcal{G})}] \\
&= \mathbb{E}_{q_\Psi(\mathbf{Z}|\mathcal{G})}[\log p_\theta(\mathbf{Z}|\mathcal{G})] - \mathrm{KL}(q_\Psi(\mathbf{Z}|\mathcal{G})||p(\mathbf{Z})).
\end{aligned} \tag{20}$$

Second, the lower bound of denoising variational inference in Eq. 6 can be derived as:

$$\begin{aligned}
\mathcal{L}_d &= \mathbb{E}_{q'_\Psi(\mathbf{Z}|\mathcal{G})}[\log \frac{p_\theta(\mathbf{Z}, \mathcal{G})}{q'_\Psi(\mathbf{Z}|\mathcal{G})}] \geq \mathbb{E}_{q'_\Psi(\mathbf{Z}|\mathcal{G})}\left[\log \frac{p_\theta(\mathcal{G}, \mathbf{Z})}{q_\Psi(\mathbf{Z}|\tilde{\mathcal{G}})}\right] \\
&= \mathbb{E}_{q'_\Psi(\mathbf{Z}|\mathcal{G})}[\log p_\theta(\mathcal{G}|\mathbf{Z}) + \log p(\mathbf{Z}) - \log q_\Psi(\mathbf{Z}|\tilde{\mathcal{G}})] \\
&= \mathbb{E}_{q'_\Psi(\mathbf{Z}|\mathcal{G})}[\log p_\theta(\mathcal{G}|\mathbf{Z})] - \mathbb{E}_{q'_\Psi(\mathbf{Z}|\mathcal{G})}\left[\log \frac{q_\Psi(\mathbf{Z}|\tilde{\mathcal{G}})}{p(\mathbf{Z})}\right] \\
&= \mathbb{E}_{q'_\Psi(\mathbf{Z}|\mathcal{G})}[\log p_\theta(\mathcal{G}|\mathbf{Z})] - \mathbb{E}_{q(\tilde{\mathcal{G}}|\mathcal{G})}\mathbb{E}_{q_\Psi(\mathbf{Z}|\mathcal{G})}\left[\log \frac{q_\Psi(\mathbf{Z}|\tilde{\mathcal{G}})}{p(\mathbf{Z})}\right] \\
&= \mathbb{E}_{q'_\Psi(\mathbf{Z}|\mathcal{G})}[\log p_\theta(\mathcal{G}|\mathbf{Z})] - \mathbb{E}_{q(\tilde{\mathcal{G}}|\mathcal{G})}[\mathrm{KL}(q_\Psi(\mathbf{Z}|\tilde{\mathcal{G}}))||p(\mathbf{Z})].
\end{aligned}$$

Third, we derive an upper bound for GIB in Eq. 13. We decompose the mutual information:

$$\mathrm{I}(\hat{y}; \mathcal{G}_s) = \mathbb{E}_{p(\hat{y}, \mathcal{G}_s)}\left[\log \frac{p(\hat{y}, \mathcal{G}_s)}{p(\hat{y})p(\mathcal{G}_s)}\right], \mathrm{I}(\mathcal{G}; \mathcal{G}_s) = \mathbb{E}_{p(\mathcal{G}, \mathcal{G}_s)}\left[\log \frac{p(\mathcal{G}, \mathcal{G}_s)}{p(\mathcal{G})p(\mathcal{G}_s)}\right]. \tag{21}$$

The GIB objective can be written as:

$$
\begin{aligned}
&- \mathrm{I}(\hat{y}; \mathcal{G}_s) + \beta \, \mathrm{I}(\mathcal{G}; \mathcal{G}_s) \\
&= -\mathbb{E}_{p(\hat{y}, \mathcal{G}_s)} \left[ \log \frac{p(\hat{y}, \mathcal{G}_s)}{p(\hat{y}) p(\mathcal{G}_s)} \right] + \beta \mathbb{E}_{p(\mathcal{G}, \mathcal{G}_s)} \left[ \log \frac{p(\mathcal{G}, \mathcal{G}_s)}{p(\mathcal{G}) p(\mathcal{G}_s)} \right] \\
&= -\mathbb{E}_{p(\hat{y}, \mathcal{G}_s)} \left[ \log \frac{p(\hat{y}|\mathcal{G}_s) p(\mathcal{G}_s)}{p(\hat{y}) p(\mathcal{G}_s)} \right] + \beta \mathbb{E}_{p(\mathcal{G}, \mathcal{G}_s)} \left[ \log \frac{p(\mathcal{G}_s|\mathcal{G}) p(\mathcal{G})}{p(\mathcal{G}) p(\mathcal{G}_s)} \right] \\
&= -\mathbb{E}_{p(\hat{y}, \mathcal{G}_s)} \left[ \log \frac{p(\hat{y}|\mathcal{G}_s)}{p(\hat{y})} \right] + \beta \mathbb{E}_{p(\mathcal{G}, \mathcal{G}_s)} \left[ \log \frac{p(\mathcal{G}_s|\mathcal{G})}{p(\mathcal{G}_s)} \right] \\
&= -\mathbb{E}_{p(\mathcal{G}_s)} \mathbb{E}_{p(\hat{y}|\mathcal{G}_s)} \left[ \log \frac{p(\hat{y}|\mathcal{G}_s)}{p(\hat{y})} \right] + \beta \mathbb{E}_{p(\mathcal{G})} \mathbb{E}_{p(\mathcal{G}_s|\mathcal{G})} \left[ \log \frac{p(\mathcal{G}_s|\mathcal{G})}{p(\mathcal{G}_s)} \right].
\end{aligned}
\tag{22}
$$

Using Jensen's inequality and assuming that $p_f(\hat{y}|\mathcal{G}_s)$ is an approximation of $p(\hat{y}|\mathcal{G}_s)$, we can get:

$$
\begin{aligned}
-\mathbb{E}_{p(\mathcal{G}_s)} \mathbb{E}_{p(\hat{y}|\mathcal{G}_s)} \left[ \log \frac{p(\hat{y}|\mathcal{G}_s)}{p(\hat{y})} \right] &\leq -\mathbb{E}_{p(\mathcal{G}_s)} \mathbb{E}_{p(\hat{y}|\mathcal{G}_s)} [\log p_f(\hat{y}|\mathcal{G}_s)] - \mathbb{E}_{p(\hat{y})} [\log p(\hat{y})] \\
&= -\mathbb{E}_{p(\mathcal{G}_s, \hat{y})} [\log p_f(\hat{y}|\mathcal{G}_s)] + \mathrm{H}(\hat{y}).
\end{aligned}
\tag{23}
$$

We introduce explain models:

$$
\begin{aligned}
&\beta \mathbb{E}_{p(\mathcal{G})} \mathbb{E}_{p(\mathcal{G}_s|\mathcal{G})} \left[ \log \frac{p(\mathcal{G}_s|\mathcal{G})}{p(\mathcal{G}_s)} \right] \\
&= \mathbb{E}_{p(\mathcal{G})} \mathbb{E}_{p(\mathcal{G}_s|\mathcal{G})} \left[ \log \frac{p_\alpha(\mathcal{G}_s|\mathcal{G})}{p(\mathcal{G}_s)} \cdot \frac{p(\mathcal{G}_s|\mathcal{G})}{p_\alpha(\mathcal{G}_s|\mathcal{G})} \right] \\
&= \mathbb{E}_{p(\mathcal{G})} \mathbb{E}_{p(\mathcal{G}_s|\mathcal{G})} \left[ \log \frac{p_\alpha(\mathcal{G}_s|\mathcal{G})}{p(\mathcal{G}_s)} \right] + \mathbb{E}_{p(\mathcal{G})} \mathbb{E}_{p(\mathcal{G}_s|\mathcal{G})} \left[ \log \frac{p(\mathcal{G}_s|\mathcal{G})}{p_\alpha(\mathcal{G}_s|\mathcal{G})} \right].
\end{aligned}
\tag{24}
$$

The second term is the KL divergence:

$$
\mathbb{E}_{p(\mathcal{G})} \mathbb{E}_{p(\mathcal{G}_s|\mathcal{G})} \left[ \log \frac{p(\mathcal{G}_s|\mathcal{G})}{p_\alpha(\mathcal{G}_s|\mathcal{G})} \right] = \mathbb{E}_{p(\mathcal{G})} [\mathrm{KL}(p(\mathcal{G}_s|\mathcal{G}) \| p_\alpha(\mathcal{G}_s|\mathcal{G}))] \geq 0.
\tag{25}
$$

Therefore,

$$
\mathrm{I}(\mathcal{G}; \mathcal{G}_s) \leq \mathbb{E}_{p(\mathcal{G})} \mathbb{E}_{p_\alpha(\mathcal{G}_s|\mathcal{G})} \left[ \log \frac{p_\alpha(\mathcal{G}_s|\mathcal{G})}{q(\mathcal{G}_s)} \right] = \mathbb{E}_{p(\mathcal{G})} [\mathrm{KL}(p_\alpha(\mathcal{G}_s|\mathcal{G}) \| q(\mathcal{G}_s))].
\tag{26}
$$

Combined with our previous derivation of the first term, we can get:

$$
- \mathrm{I}(\hat{y}; \mathcal{G}_s) + \beta \, \mathrm{I}(\mathcal{G}; \mathcal{G}_s) \leq -\mathbb{E}_{p(\mathcal{G}_s, \hat{y})} \left[ \log p_f(\hat{y}|\mathcal{G}_s) \right] + \mathrm{H}(\hat{y}) + \beta \mathbb{E}_{p(\mathcal{G})} \left[ \mathrm{KL}(p_\alpha(\mathcal{G}_s|\mathcal{G}) \| q(\mathcal{G}_s)) \right].
\tag{27}
$$

# D  EXPERIMENT SUPPLEMENT

Table 2: Statistics of Datasets.

| Dataset | DBLP | ACM | Freebase |
|---|---|---|---|
| Nodes | 26,128 | 10942 | 43,854 |
| Edges | 239,566 | 547872 | 151,034 |
| Node Types | 4 | 4 | 4 |
| Edge Types | 6 | 8 | 6 |
| Classes | 4 | 3 | 3 |

## D.1  DATASETS

We conduct experiments on three real-world datasets. According to the Heterogeneous Graph Benchmark (Lv et al., 2021) settings, we randomly split the nodes with proportions of 24%, 6%, and 70% for training, validation, and testing, respectively. The statistics of the three datasets are shown in Table 2.

- **DBLP**[1] is a computer science bibliography network that contains four types of nodes: Paper (P), Author (A), Term (T), and Venue (V). The authors in this network are from four research areas (*Database, Data Mining, Artificial Intelligence,* and *Information Retrieval*).
- **ACM**[2] is a citation network that contains four types of nodes: Paper (P), Author (A), Term (T), and Subject (S). The papers in this network are divided into three classes (*Database, Wireless Communication,* and *Data Mining*).
- **Freebase** (Bollacker et al., 2008) is a knowledge graph that contains four types of nodes: Movie (M), Actor (A), Director (D) and Writer (W).

## D.2 BASELINES

Next, we provide details on the baselines used in our experiments.

- **PGExpaliner** (Luo et al., 2020) is a parameterized explainer that learns a mask for each edge to obtain edge importance scores.
- **GNNExplainer** (Ying et al., 2019) maximizes the mutual information between the model's prediction on the original input and the masked input by masking features and edges.
- **PGM-Explainer** (Vu & Thai, 2020) employs a Bayesian network-based approach, treating vertices in the input graph as random variables to fit the GNN model's predicted label.
- **V-InfoR** (Wang et al., 2024) utilizes a parametric method, learning edge masks on the latent representations to identify important edges.
- **PGE-Relation** is an extension of PGExplainer, where we replace the initial concatenation with a relation-based attention learning module to enable learning of heterogeneous semantics.

## D.3 BASE HETEROGENEOUS GRAPH NEURAL NETWORK

Table 3: Node classification result using our heterogeneous Graph Neural Network.

| Dataset | DBLP | ACM | Freebase |
|---|---|---|---|
| Micro-F1 | 92.64±0.14 | 92.32±0.12 | 65.93±0.20 |
| Macro-F1 | 92.16±0.19 | 92.40±0.11 | 61.94±0.36 |

In the experiment, we use a basic heterogeneous Graph Neural Network, which encodes the input graph through 2 layers of GCN, and then used a layer of attention learning module to learn different heterogeneous relations. For a heterogeneous graph, the feature spaces of different types of nodes are usually different. We use a mapping function to map the features of different types into a common feature space, as shown below:

$$\mathbf{z}_v = \mathbf{W}_m \mathbf{x}_v^A + \mathbf{b}_m, \tag{28}$$

where $A \in \mathcal{A}$ is the node type of node $v$, $\mathbf{W}_m$ is a learnable weight, and $\mathbf{b}_m$ is the bias. Then, in the shared space, we use GCN to obtain the node embeddings:

$$\mathbf{Z}^{(l)} = \text{GCN}(\mathbf{Z}^{(l-1)}, \mathbf{A}), \mathbf{Z}^{(0)} = \mathbf{Z}_v. \tag{29}$$

To learn the heterogeneous semantics of the heterogeneous graph, we introduce a type vector $\boldsymbol{\gamma}_v$ and learn relation information through an attention module:

$$\boldsymbol{\gamma}_i^q = \mathbf{W}_r^q \boldsymbol{\gamma}_i, \boldsymbol{\gamma}_j^k = \mathbf{W}_r^k \boldsymbol{\gamma}_j,$$
$$score_{ij}^{\gamma} = \boldsymbol{\gamma}_i^q \boldsymbol{\gamma}_j^k, \tag{30}$$

where $\mathbf{W}_r^q$ and $\mathbf{W}_r^k$ are learnable weights. The attention of the nodes can be computed as follows:

$$q_i = \mathbf{W}_q^z \mathbf{z}_i, k_j = \mathbf{W}_k^z \mathbf{z}_j,$$
$$\widehat{\alpha}_{ij} = \frac{\exp(\text{LeakyReLU}(a^T[q_i \parallel k_j]))}{\sum_{j' \in \mathcal{N}_i} \exp(\text{LeakyReLU}(a^T[q_i \parallel k_{j'}]))}. \tag{31}$$

---

[1]https://dblp.uni-trier.de

[2]http://dl.acm.org/

where $\mathbf{W}_q^z$ and $\mathbf{W}_k^z$ are learnable weights. The final prediction can be expressed as:

$$score_{ij} = \widehat{\alpha}_{ij} + \beta score_{ij}^{\gamma},$$
$$\mathbf{Z}_{\mathbf{H}}^{(l)} = \text{LayerNorm}(\mathbf{Z}_{\mathbf{H}}^{(l-1)} + score_{ij} \cdot \mathbf{Z}_{\mathbf{H}}^{(l-1)}), \tag{32}$$
$$\hat{y} = P_f(\mathbf{Z}_{\mathbf{H}}^{(l)}; \theta_p).$$

where $\theta_p$ is the parameter of the predictor. The basic prediction results are shown in Table 3.

The experiments are conducted on an L20 GPU with 48GB of memory. Our CPU is an Intel(R) Xeon(R) Platinum 8457C. We utilized PyTorch version 1.13.1 and DGL version 1.1.1.

## D.4 VARIANCE

In this section, we report the variance results for the comparison of RHGIB and baselines under different ratios of random structural noise, which serves as a supplement to Table 1. Table 4 presents the variance of the results under different noise ratios.

Table 4: The variance of the results under different noise ratios.

| Dataset | Noise Ratio | 10% | | 20% | | 30% | | 40% | |
|---|---|---|---|---|---|---|---|---|---|
| | | MAE-Var | RMSE-Var | MAE-Var | RMSE-Var | MAE-Var | RMSE-Var | MAE-Var | RMSE-Var |
| DBLP | PGExplainer | 0.0062 | 0.0054 | 0.0089 | 0.0666 | 0.0059 | 0.0024 | 0.0068 | 0.0042 |
| | GNNExplainer | 0.0009 | 0.0006 | 0.0007 | 0.0004 | 0.0011 | 0.0007 | 0.0008 | 0.0004 |
| | PGM-Explainer | 0.0007 | 0.0005 | 0.0006 | 0.0003 | 0.0002 | 0.0001 | 0.0005 | 0.0001 |
| | V-InfoR | 0.0030 | 0.0027 | 0.0025 | 0.0020 | 0.0026 | 0.0020 | 0.0010 | 0.0008 |
| | PGE-Relation | 0.0008 | 0.0004 | 0.0005 | 0.0002 | 0.0006 | 0.0003 | 0.0006 | 0.0004 |
| | RHGIB | 0.0029 | 0.0017 | 0.0018 | 0.0013 | 0.0034 | 0.0022 | 0.0015 | 0.0008 |
| ACM | PGExplainer | 0.0080 | 0.0037 | 0.0162 | 0.0069 | 0.0152 | 0.0127 | 0.0181 | 0.0124 |
| | GNNExplainer | 0.0003 | 0.0005 | 0.0001 | 0.0002 | 0.0003 | 0.0001 | 0.0002 | 0.0002 |
| | PGM-Explainer | 0.0009 | 0.0004 | 0.0003 | 0.0002 | 0.0006 | 0.0004 | 0.0007 | 0.0004 |
| | V-InfoR | 0.0001 | 0.0002 | 0.0004 | 0.0004 | 0.0003 | 0.0007 | 0.0005 | 0.0003 |
| | PGE-Relation | 0.0003 | 0.0001 | 0.0005 | 0.0004 | 0.0002 | 0.0001 | 0.0001 | 0.0001 |
| | RHGIB | 0.0009 | 0.0012 | 0.0010 | 0.0008 | 0.0019 | 0.0013 | 0.0015 | 0.0008 |
| Freebase | PGExplainer | 0.0096 | 0.0071 | 0.0078 | 0.0051 | 0.0041 | 0.0039 | 0.0019 | 0.0018 |
| | GNNExplainer | 0.0002 | 0.0001 | 0.0003 | 0.0002 | 0.0003 | 0.0002 | 0.0007 | 0.0005 |
| | PGM-Explainer | 0.0003 | 0.0001 | 0.0001 | 0.0001 | 0.0004 | 0.0002 | 0.0009 | 0.0004 |
| | V-InfoR | 0.0322 | 0.0233 | 0.0527 | 0.0566 | 0.0329 | 0.0277 | 0.0111 | 0.0054 |
| | PGE-Relation | 0.0007 | 0.0003 | 0.0005 | 0.0004 | 0.0003 | 0.0002 | 0.0001 | 0.0001 |
| | RHGIB | 0.0010 | 0.0018 | 0.0012 | 0.0010 | 0.0021 | 0.0012 | 0.0014 | 0.0007 |

## D.5 PARAMETER SETTING

For the base heterogeneous Graph Neural Network, we use Adam (Kingma & Ba, 2014) as the optimizer, LeakyReLU with a negative slope $s = 0.2$ as the activation function, a learning rate of 1e-4, and a dropout rate of 0 for Freebase and 0.5 for other datasets. The hidden dimension is set to 256. For RHGIB, we use Adam as the optimizer with a learning rate of 1e-4. We set the hidden dimension for variational inference to 64, the output dimension to 32, and the edge weight output dimension to 32. Our training is performed for 100 epochs.

