# OpenReview forum: "Robust Heterogeneous Graph Neural Network Explainer with Graph Information Bottleneck"
_ICLR.cc/2025/Conference — Submitted to ICLR 2025_

### Official Review · Reviewer_oe9w · 2024-10-17

**Soundness:** 3
**Presentation:** 2
**Contribution:** 2
**Rating:** 3
**Confidence:** 5

**Summary:**

This paper proposes a robust heterogeneous GNN explainer based on the graph information bottleneck (RHGIB) to identify the critical subgraph which decides the prediction. RHGIB consists of a denoising variational inference process and a relation-aware explanation generator. The denoising variational inference deduces the statistic characteristics of the graphs and provides the robust edge representations. The relation-aware explanation generator introduces the heterogeneous edge type to constrains the explanation subgraph optimization. Experiments on three public heterogeneous graph datasets demonstrate the superiority of RHGIB. The ablation study and the fidelity-sparsity analysis further present insights of the proposed method.

**Strengths:**

1. This work provides solid proofs to the denoising variational inference process and the GIB optimization, and the
theorem of heterogeneous graph noise amplification effect is meaningful.
2. The focus of explanation on heterogeneous graphs and heterogeneous GNNs is intriguing and practical.
3. The superiority of RHGIB is sufficiently demonstrated by the extensive experiments.

**Weaknesses:**

1. The contribution of this work needs more elaborations, since the framework of RHGIB is highly similar to the baseline V-InfoR, consisting of a robust feature encoder and a GIB-based explainer.
2. The manuscript overlooks the main distinction of the proposed denoising variational inference against the standard variational inference and the target of Formula (4) needs more explanations from my perspective.
3. The case study in Section 5.6 is not convincing. The significance of <Paper-5558> to predict <Author-1034> is opaque and the same issue persists in the second example .
4. The writing and organization of this manuscript can be further polished up. Moreover, Figure 1 in introduction is too rough and lacks of insights.

**Questions:**

1. If the edge representations are directly sampled from the variational distribution? The forward process deserves detailed introduction.
2. Elaboration on the denoising variational inference is recommended, especially for the motivation of Formula (4).
3. How to acquire the edge type embedding $r_{\phi(e)}$ is missing in the manuscript.
4. The reasonability of adopting MAE and RMSE to evaluate the classification task should be clarified.

---

> ### Author Response · Authors · 2024-11-27
> **Responses to Reviewer oe9w**
>
> Thank you for your review and your comments. We hope that our answers below help resolve any questions or potential misunderstandings.
>
> >1. If the edge representations are directly sampled from the variational distribution? The forward process deserves detailed introduction.
>
> In the variational inference, the process of sampling edge representations directly from the variational distribution is part of the generative process for reconstructing the graph structure:
>
> 1. RHGIB uses an encoder (GCN) to learn latent representations for each node in the graph. The encoder outputs the mean and variance of the variational distribution for each node's latent representation, typically assuming a Gaussian distribution for the latent variables.
>
> 2. Once the encoder outputs the mean and variance for each node, we sample the latent node embeddings from a Gaussian distribution $\mathcal{N}(\mu_i, \sigma_i^2)$ for each node $v_i$, where $\mu_i$ and $\sigma_i^2$ are the mean and variance produced by the encoder for node $v_i$.
>
> 3. The goal is to predict the edges in the graph, and the edges are assumed to be generated from the dot product of the latent node representations. Specifically, the edge between two nodes $v_i$ and $v_j$ is predicted as follows: $\hat{A}_{ij} = \sigma(z_i^T z_j)$ where $z_i$ and $z_j$ are the latent embeddings of nodes $v_i$ and $v_j$, and $\sigma(\cdot)$ is a sigmoid function that squashes the result into a probability (between 0 and 1).
>
> 4. For each node, sample latent variables $z_i$ from the variational distribution $\mathcal{N}(\mu_i, \sigma_i^2)$, and then compute the edge predictions based on these sampled values.
>
> 5. To allow backpropagation through the sampling process (since sampling is non-differentiable), the reparameterization trick is used. The reparameterization trick expresses the latent variable $z_i$ as: $z_i = \mu_i + \sigma_i \cdot \epsilon$ where $\epsilon \sim \mathcal{N}(0, I)$ is a random noise vector sampled from a standard normal distribution, and $\mu_i$ and $\sigma_i$ are the mean and standard deviation predicted by the encoder.
>
> >2. Elaboration on the denoising variational inference is recommended, especially for the motivation of Formula (4).
>
> The motivation for performing denoising variational inference lies in the fact that heterogeneous graphs in the real world are often accompanied by noise, which can negatively affect the predictions of graph neural networks and lead to biased explanation results. By using denoising variational inference, we can effectively extract clean node representations from the original features and graph structure, mitigating the impact of noise and ensuring that the model's predictions are more accurate and reliable. This also enhances the model's robustness and its ability to understand complex semantics. At the same time, we have theoretically proven that existing heterogeneous graph methods amplify the effects of noise, further emphasizing the need to take measures to mitigate information degradation caused by noise. In Equation (4), due to the different distributions between noisy graph data and standard graph data, we transform the noisy distribution into the original distribution from a distributional perspective, introducing a transformation of the distribution.
>
> >3. How to acquire the edge type embedding $\mathbf{r}_{\varphi(e)}$ is missing in the manuscript.
>
> We apologize for any confusion caused by our wording. We directly use one-hot encoding to encode each edge type, which is a common operation in other heterogeneous graph methods, such as simple-HGN [1], HINormer [2], and PHGT [3]. We have included the relevant details in the revised manuscript.
>
> Reference:
>
> [1] Lv, Qingsong, et al. "Are we really making much progress? revisiting, benchmarking and refining heterogeneous graph neural networks." *Proceedings of the 27th ACM SIGKDD conference on knowledge discovery & data mining*. 2021.
>
> [2] Mao, Qiheng, et al. "Hinormer: Representation learning on heterogeneous information networks with graph transformer." Proceedings of the ACM Web Conference 2023. 2023.
>
> [3] Lu, Zhiyuan, et al. "Heterogeneous Graph Transformer with Poly-Tokenization." *Proceedings of the Thirty-Third International Joint Conference on Artificial Intelligence, IJCAI-24, International Joint Conferences on Artificial Intelligence Organization*. 2024.

---

> > ### Author Response · Authors · 2024-11-27
> > **Responses to Reviewer oe9w**
> >
> > >4. The reasonability of adopting MAE and RMSE to evaluate the classification task should be clarified.
> >
> > MAE and RMSE are metrics used to evaluate the performance of the explainer. (MAE, $\frac{1}{N}\sum _ {i=1}^{N}\left|\mathbb{I}(\hat{y} _ {i}=y _ {i})-\mathbb{I}(\hat{y} _ {i}^{\mathcal{G} _ s}=y _ {i})\right|$) and  (RMSE, $\sqrt{\frac1N\sum _ {i=1}^N(\mathbb{I}(\hat{y} _ {i}=y _ {i})-\mathbb{I}(\hat{y} _ {i}^{\mathcal{G} _ s}=y _ {i}))^2}$) effectively assess the performance variation between the key subgraphs obtained by the explainer and the original graph. The indicator function $\mathbb{I}(\cdot)$ is used to indicate whether the model's outputs at different stages match the predictions of the original graph. A smaller MAE indicates that the predictions generated by the explainer are closer to the original model's predictions. RMSE evaluates the performance of the explainer by comparing the magnitude of prediction errors; larger errors have a greater impact on the RMSE.

---

> > > ### Author Response · Authors · 2024-12-03
> > >
> > > We have answered all the reviewer's concerns in the above comment (with new experiments) and the rebuttal. We request the reviewer to please take a look and kindly consider increasing their score.

---

> > > > ### Comment · Reviewer_oe9w · 2024-12-03
> > > >
> > > > Thanks for your detailed responses. The main concerns still exist, therefore I decide to keep my original mark.

---

### Official Review · Reviewer_dnyQ · 2024-11-03

**Soundness:** 1
**Presentation:** 2
**Contribution:** 2
**Rating:** 3
**Confidence:** 4

**Summary:**

This paper presents a PHGIB model aimed at mitigating the impact of noise on the graph explanation results. Specifically, the authors theoretically demonstrate how noise affects the information propagation process in heterogeneous graphs. Furthermore, they utilize noise-variational inference to alleviate its effects and combine these results with a graph information bottleneck to optimize the heterogeneous subgraph generation process for robust explanations.

**Strengths:**

S1: This paper primarily addresses the impact of noise in real-world heterogeneous graphs on graph interpretability. The model proposed by the authors features a unique denoising variational graph encoder, which can effectively integrate denoising strategies into a variational inference framework. Additionally, the authors combine edge-type information with an attention mechanism used during generation; this can enhance the capture of semantic information in heterogeneous graphs.

S2: The proposed method is supported by a theoretical foundation, which is based on validated techniques, and a theoretical analysis of the improved method is provided. The authors also present relevant experiments to validate their approach.

S3: The organization of the paper is well-structured, and the description of the methods is logically coherent.

**Weaknesses:**

W1. The authors have not conducted a comprehensive review of related works. Although the paper states that there are no existing robust heterogeneous explainers, it introduces a related method and claims to validate the robustness of the model. However, similar studies, such as "Counterfactual Learning on Heterogeneous Graphs with Greedy Perturbation" and "On the Robustness of Post-hoc GNN Explainers to Label Noise," are not mentioned. Furthermore, the authors do not discuss other papers related to heterogeneous graph explainers or clarify the differences between their method and existing ones, which limits the demonstration of the originality of their work.

W2. The authors only utilize MAE and RMSE as metrics to evaluate the interpretability of the extracted subgraphs, while related literature typically includes additional classification-related evaluation metrics to assess model performance comprehensively.

W3. In the baseline design, the authors only select general graph explanation methods and do not include interpretable methods specifically for heterogeneous graphs, which fails to effectively demonstrate the advantages of their work.

W4. Although the authors provide visualization results in the case study to demonstrate the effectiveness of their explanations, they do not present results from other explainers, which diminishes the credibility and superiority of their proposed model.

W5. The experiments described in the paper are based on artificially added noise values, which only demonstrate some robustness of the model but do not validate its generalization capability. Furthermore, these noise values are limited to edge perturbations, while the introduction also states that node features can be distorted by noise; thus, the experiments should consider including noise that accounts for this assumption.

**Questions:**

Q1. How can it be proven that the statistical features and distributions of graph data are affected by existing noise? Since the main goal of the paper is to eliminate noise present in real-world data, the authors only provide results where noise has been added to edges in the real dataset. How can these experimental results effectively reflect the noise in the real world? Is it possible to elaborate on the specific impacts of noise on the statistical features of graph data and provide related experiments or theoretical support to demonstrate the model's ability to effectively remove noise from real-world data?

Q2. In Proof of THEOREM 1, why the method based on neighbor aggregation does not consider the influence of node v_j's neighbors on node v_i? This could affect the effectiveness of neighbor aggregation, and it is suggested that the authors provide a detailed explanation of the rationale behind this assumption and discuss the interactions between nodes during the aggregation process.

Q3. Regarding hyperparameter analysis, the Lagrange multiplier is used to control the extent of graph information compression. As stated in the paper, studies on "Graph Structure Learning with Variational Information Bottleneck" indicate that excessive or insufficient compression should degrade the model's performance. However, the authors' experimental results show that RHGIB is insensitive to these constraints, raising concerns about whether the model adequately captures patterns in the data. It is recommended that the authors carefully examine the model architecture and hyperparameter settings to further validate the model's effectiveness and robustness, ensuring that it can fully leverage the data's features.

---

> ### Author Response · Authors · 2024-11-27
> **Responses to Reviewer dnyQ**
>
> Thank you for your review and your comments. We hope that our answers below help resolve any questions or potential misunderstandings.
>
> >Q1. How can it be proven that the statistical features and distributions of graph data are affected by existing noise? Since the main goal of the paper is to eliminate noise present in real-world data, the authors only provide results where noise has been added to edges in the real dataset. How can these experimental results effectively reflect the noise in the real world? Is it possible to elaborate on the specific impacts of noise on the statistical features of graph data and provide related experiments or theoretical support to demonstrate the model's ability to effectively remove noise from real-world data?
>
> Noise in real-world data is difficult to identify, so it is typically added manually. This is also the approach taken in related fields, such as adversarial attacks [1,2,3], for experimentation. Many existing studies have explored noise in the real world, such as applications in healthcare [4], communication systems [5], and image processing [6], demonstrating that our random noise can reflect real-world scenarios.
>
> We measure the impact of noise on the graph distribution through the variation of the KL divergence, and it can be observed that noise significantly disrupts the graph distribution:
>
> | noise ratio | 10%     | 20%     | 30%     | 40%     | 50%     | 60%     | 70%     | 80%     | 90%     |
> | ----------- | ------- | ------- | ------- | ------- | ------- | ------- | ------- | ------- | ------- |
> | KL          | 0.11108 | 0.31823 | 0.54854 | 0.80690 | 1.09782 | 1.43823 | 2.01607 | 2.15205 | 2.40943 |
>
> We evaluated the denoising performance under a 60% noise scenario (which significantly exacerbates the damage to the graph), and the comparison results with the ground truth are as follows:
>
> | Dataset | DBLP   | ACM    | Freebase |
> | ------- | ------ | ------ | -------- |
> | ACC.(%) | 79.98% | 74.77% | 77.53%   |
>
> Reference:
>
> [1] Zhu, Dingyuan, et al. "Robust graph convolutional networks against adversarial attacks." *Proceedings of the 25th ACM SIGKDD international conference on knowledge discovery & data mining*. 2019.
>
> [2] Zhang, Mengmei, et al. "Robust heterogeneous graph neural networks against adversarial attacks." *Proceedings of the AAAI Conference on Artificial Intelligence*. Vol. 36. No. 4. 2022.
>
> [3] Ma, Jiaqi, Shuangrui Ding, and Qiaozhu Mei. "Towards more practical adversarial attacks on graph neural networks." *Advances in neural information processing systems* 33 (2020): 4756-4766.
>
> [4] Gravel, Pierre, Gilles Beaudoin, and Jacques A. De Guise. "A method for modeling noise in medical images." *IEEE Transactions on medical imaging* 23.10 (2004): 1221-1232.
>
> [5] Farsad, Nariman, et al. "Channel and noise models for nonlinear molecular communication systems." *IEEE Journal on Selected Areas in Communications* 32.12 (2014): 2392-2401.
>
> [6] George, Swapna Mol, and P. Muhamed Ilyas. "A review on speech emotion recognition: a survey, recent advances, challenges, and the influence of noise." *Neurocomputing* 568 (2024): 127015.
>
> >Q2. In Proof of THEOREM 1, why the method based on neighbor aggregation does not consider the influence of node v_j's neighbors on node v_i? This could affect the effectiveness of neighbor aggregation, and it is suggested that the authors provide a detailed explanation of the rationale behind this assumption and discuss the interactions between nodes during the aggregation process.
>
> The meta-path-based method directly aggregates all nodes along the meta-path, considering multi-hop nodes simultaneously based on the length of the meta-path. As a result, the neighbors of node $v_j$ have a significant impact on node $v_i$, which is why we found that this approach can amplify the noise effect. In contrast, the neighbor aggregation-based method, within the message-passing framework, only considers the influence of node $v_i$'s neighbors at each layer, and only at the second layer does it begin to incorporate the information from the neighbors of node $v_j$. Moreover, the influence is inversely related to the degree, making it much smaller compared to the meta-path-based method. Additionally, during the second layer of aggregation, the meta-path-based method further expands its influence range, dramatically amplifying the noise effect.

---

> > ### Author Response · Authors · 2024-11-27
> > **Responses to Reviewer dnyQ**
> >
> > >Q3. Regarding hyperparameter analysis, the Lagrange multiplier is used to control the extent of graph information compression. As stated in the paper, studies on "Graph Structure Learning with Variational Information Bottleneck" indicate that excessive or insufficient compression should degrade the model's performance. However, the authors' experimental results show that RHGIB is insensitive to these constraints, raising concerns about whether the model adequately captures patterns in the data. It is recommended that the authors carefully examine the model architecture and hyperparameter settings to further validate the model's effectiveness and robustness, ensuring that it can fully leverage the data's features.
> >
> > We re-examined the model architecture and hyperparameter settings and found that this is due to the fact that the MAE and RMSE of our model remove smaller scores in both parts (↓ is good). In this case, the degree of variation visually appears small in the image, but it is still consistent with the changes in the Lagrange multiplier on the actual data.

---

> > ### Author Response · Authors · 2024-12-03
> >
> > We have answered all the reviewer's concerns in the above comment (with new experiments) and the rebuttal. We request the reviewer to please take a look and kindly consider increasing their score.

---

> > > ### Comment · Reviewer_dnyQ · 2024-12-03
> > >
> > > I appreciate the author's responses. However, I don't think my major concerns have been addressed.

---

### Official Review · Reviewer_mA7h · 2024-11-03

**Soundness:** 3
**Presentation:** 2
**Contribution:** 3
**Rating:** 3
**Confidence:** 4

**Summary:**

This paper proposes RHGIB, to address the challenges of explaining heterogeneous graph neural networks with noise. The authors theoretically analyzed the amplification effect of noise in HG, particularly meta-path-based methods; incorporated heterogeneous edge types into the generation of subgraphs and utilize the GIB for optimization. Experiments were conducted on real HGs.

**Strengths:**

S1: Motivation: there are noise in HG when building the graph. How to explain GNN effectiveness on HG with noise worth to study

S2:  Theoretical analyses the amplification effect of noise in HG with meta-path-based methods

S3: Utilizes the Graph Information Bottleneck framework for optimization, enabling the explainer to handle irregularities introduced by structural noise.

**Weaknesses:**

C1: For graph learning with HG, meta-path-based algorithms in one of the key technique. But how to select meta-path is important. Meta-paths are usually given by domain experts. As shown in the methdology and theoretical analyses, the proposed is robust to noise with meta-path-based methods. Would the theorem be applied for non-meta-path methods? For example, for node classification, a simple GCN/GAT on the target node layer/domain without meta-path is a straightforward baseline. How is the effect of noise for a simple GNN without meta-path?

C2: The methodology section could be more concise and easier to follow.

C3: How does the proposed handle feature noise in addition to structural noise? The paper seems to focus primarily on structural noise.

C4: The paper does not explore the generalizability of RHGIB to other graph tasks except for node classification, such as link prediction or graph classification as mentioned in introduction.

C5: The main contribution of this paper is on robustness on HG with noise. However, there is no experiments on specific investigation of the robustness. For example, a synthetic HG with community structure can be generated with existing graph generator (groundtruth is know for the synthetic graphs), such as  Stochastic Block Model (SBM), Girvan-Newman Algorithm, and Caveman Graph Generator. Noise can be added by random deleting/adding edges. How would the proposed explain a GNN node clustering ability for the two HGs (one is noise-free, the other is with noise)?

**Questions:**

Please see weaknesses

---

> ### Author Response · Authors · 2024-11-27
> **Responses to Reviewer mA7h**
>
> Thank you for your review and your comments. We hope that our answers below help resolve any questions or potential misunderstandings.
>
> >C1: For graph learning with HG, meta-path-based algorithms in one of the key technique. But how to select meta-path is important. Meta-paths are usually given by domain experts. As shown in the methdology and theoretical analyses, the proposed is robust to noise with meta-path-based methods. Would the theorem be applied for non-meta-path methods? For example, for node classification, a simple GCN/GAT on the target node layer/domain without meta-path is a straightforward baseline. How is the effect of noise for a simple GNN without meta-path?
>
> I believe you may have misunderstood our approach. We have theoretically demonstrated the amplification effect of noise in heterogeneous graph neural networks using meta-path-based methods. In Appendix B, we compare the impact of noise on methods without meta-paths and those based on meta-paths, which lays the foundation for our choice of the base prediction model. The architecture of the prediction model we selected is detailed in Appendix D.3.
>
> >C2: The methodology section could be more concise and easier to follow.
>
> Thank you for your comments. We have added descriptions of relevant concepts and removed unnecessary words. Additionally, we have adjusted the wording in the explanation subgraph sampling part to improve the readability of the paper.
>
> >C3: How does the proposed handle feature noise in addition to structural noise? The paper seems to focus primarily on structural noise.
>
> In this paper, we mainly focus on structural noise because it has more practical significance, such as adversarial attacks [1], structural damage [2], etc. Current research indicates that graph structural information is crucial for classification tasks [3]. Therefore, we focus on structural noise. At the same time, in the datasets we used, only the target-type nodes have features, while the features of other types of nodes are filled with a matrix of ones. We will investigate feature noise and the scenario of attribute-free graphs in our future work.
>
> Reference:
>
> [1] Liu, Zihan, et al. "Towards reasonable budget allocation in untargeted graph structure attacks via gradient debias." *arXiv preprint arXiv:2304.00010* (2023).
>
> [2] Wang, Senzhang, et al. "V-InFoR: a robust graph neural networks explainer for structurally corrupted graphs." *Advances in Neural Information Processing Systems* 36 (2024).
>
> [3] Luo, Dongsheng, et al. "Parameterized explainer for graph neural network." *Advances in neural information processing systems* 33 (2020): 19620-19631.
>
> >C4: The paper does not explore the generalizability of RHGIB to other graph tasks except for node classification, such as link prediction or graph classification as mentioned in introduction.
>
> Our method holds great potential for providing explanations in link prediction and graph classification tasks. For link prediction, please extract the subgraph around the link's k-hop (where k is the number of layers in the HGNN) and then use a relation-based explanation generator to identify the key graph components that influence the link prediction. For graph classification, you can simply add a readout layer after the HGNN to generate an explanation subgraph that has a significant impact on the overall graph classification. Our method is flexible and can provide explanations for different tasks.
>
> >C5: The main contribution of this paper is on robustness on HG with noise. However, there is no experiments on specific investigation of the robustness. For example, a synthetic HG with community structure can be generated with existing graph generator (groundtruth is know for the synthetic graphs), such as Stochastic Block Model (SBM), Girvan-Newman Algorithm, and Caveman Graph Generator. Noise can be added by random deleting/adding edges. How would the proposed explain a GNN node clustering ability for the two HGs (one is noise-free, the other is with noise)?
>
> We presented the model's performance under different levels of noise influence in Table 1, where it can be observed that RHGIB demonstrates strong robustness. The method you suggested for conducting experiments on synthetic graphs is excellent, and we will implement it in our future work. As for node clustering explanations under noisy conditions, we can first mitigate the noise through denoising variational inference, and then, during the explanation generation process, sample the key subgraphs that influence the model's decision, just as we would with a noise-free graph.

---

> > ### Author Response · Authors · 2024-12-03
> >
> > We have answered all the reviewer's concerns in the above comment and the rebuttal. We request the reviewer to please take a look and kindly consider increasing their score.

---

> > ### Comment · Reviewer_mA7h · 2024-12-03
> > **Thx for the response.**
> >
> > I appreciate the author’s response. I don’t think it addresses my concerns. I keep my original scores.

---

### Official Review · Reviewer_2RQz · 2024-11-04

**Soundness:** 2
**Presentation:** 2
**Contribution:** 2
**Rating:** 3
**Confidence:** 5

**Summary:**

This paper proposes a robust heterogeneous graph neural network explainer with graph information bottleneck (RHGIB), which infers the latent distribution of graph structure and features to alleviate the influence of noises, and utilizes the graph information bottleneck to find explanatory subgraph. The authors also theoretically analyze the power of different heterogeneous GNN architectures on the propagation of noise information. Experimental results show the effectiveness of the proposed explainer.

**Strengths:**

1. The problem studied is interesting and novel. There is no existing work on explaining heterogeneous graph neural networks.

2. Experimental results demonstrate the effectiveness of the proposed method

3. The paper has a theoretical analysis on the noise amplification effect in heterogenous graphs.

**Weaknesses:**

1. The motivation of doing graph denoising for post-hoc explanation is unclear and unreasonable to me. The post-hoc explainer should truly reflect the important subgraph structure and features that the target GNN relies on for prediction. If noisy edges play an important role in the target GNN’s prediction, the post-hoc explainer should reflect that, which is one purpose of doing post-hoc explanation, i.e., discovering potential issues in the algorithm and data. Doing denoising from the post-hoc explainer perspective would make the explainer unable to explain the target GNN well, especially when the target GNN relies on noisy edges for prediction. I think developing a robust self-explainable heterogenous GNN would be more convincing.

2. Many important details of the methodology about heterogeneous graph neural networks are missing. For example, how do the authors model $p(G|\mathbf{Z})$ for the heterogeneous graph that takes different types and edges and nodes into consideration? How do the authors model $p(\tilde{G|G}$. Currently, the whole methodology is described in a way that almost has nothing to do with heterogeneous graphs.

3. The authors only chose the most basic HGNN architecture which only contains GCN are relational learning modules as the base model. It is unclear why the authors do not adopt more HGNN architectures, especially those meta-path based HGNN, which seems to suffer more from noises. To demonstrate that the proposed explainer is flexible to explain various HGNNs, the authors should conduct experiments on various HGNNs.

4. The related work section should include heterogeneous graph neural networks and more representative and state-of-the-art GNN explainers.

5. In the experiment, it is unclear how the noises are added. Is it adding in the training set, test set or both?

6. The technical novelty is incremental. The idea of using graph information bottleneck for GNN explainer has been studied and the idea of using variational autoencoder for graph denoising is also explored. The paper just did a simple combination of two existing methods.

**Questions:**

Please see weaknesses.

---

> ### Author Response · Authors · 2024-11-27
> **Responses to Reviewer 2RQz**
>
> Thank you for your review and your comments. We hope that our answers below help resolve any questions or potential misunderstandings.
> >1. The motivation of doing graph denoising for post-hoc explanation is unclear and unreasonable to me. The post-hoc explainer should truly reflect the important subgraph structure and features that the target GNN relies on for prediction. If noisy edges play an important role in the target GNN’s prediction, the post-hoc explainer should reflect that, which is one purpose of doing post-hoc explanation, i.e., discovering potential issues in the algorithm and data. Doing denoising from the post-hoc explainer perspective would make the explainer unable to explain the target GNN well, especially when the target GNN relies on noisy edges for prediction. I think developing a robust self-explainable heterogenous GNN would be more convincing.
>
> Many studies have shown that existing graphs contain noise, which can affect model predictions, such as [1], [2], and [3]. However, there are currently no relevant explainers that interpret the model's decision-making process in the presence of noise. From our experiments, we found that existing explainers are highly susceptible to noise, and the impact is significant.** The post-hoc explainer we designed accurately reflects the key subgraph structures and features upon which the prediction relies. The relation-based explanation generator captures key components of the message-passing process through a weighted sampling method, which can be quantitatively assessed. As demonstrated by the experimental results and case studies, our approach achieves competitive performance. We argue that, regardless of whether noisy edges impact the prediction, they should be removed and not fed into the GNN for prediction, as noisy edges carry no meaningful information. Providing explanations for clean graphs (i.e., denoised graphs) is a more reasonable approach. To address potential concerns, we conducted experiments under 60% noise conditions to validate the success rate of noise elimination. This scenario significantly damages the graph features, and the denoised graph was compared with the ground truth. The experimental results are as follows:
>
> | Dataset | DBLP   | ACM    | Freebase |
> | ------- | ------ | ------ | -------- |
> | Acc.(%) | 79.98% | 74.77% | 77.53%   |
>
> The denoising module we designed effectively eliminates the interference of noisy edges. Additionally, in Section 4.1, we theoretically demonstrate the model's denoising capability. Inputting the denoised graph into the model yields the best explanation results.
>
> >2. Many important details of the methodology about heterogeneous graph neural networks are missing. For example, how do the authors model $p(\mathcal{G}|\mathbf{Z})$ for the heterogeneous graph that takes different types and edges and nodes into consideration? How do the authors model $p(\mathcal{\tilde{G}}|\mathcal{G})$. Currently, the whole methodology is described in a way that almost has nothing to do with heterogeneous graphs.
>
> We have provided further details of our method throughout the paper. In the revised manuscript, we have added an introduction to the relevant parameters, including heterogeneous attention coefficients and edge type embeddings. We also improved the description of the method to enhance readability. In the denoising inference section, we did not consider the heterogeneity of the graph because noisy edges significantly disrupt the heterogeneity, introducing spurious heterogeneity. Moreover, the false noise edges between different types of nodes generate misleading semantics for the model, and modeling the heterogeneity would only amplify the noise's impact. If heterogeneity were modeled in the $p(\mathcal{G}|\mathbf{Z})$ part, the model's inference would be interfered with by the false edge types, leading to biased predictions. In contrast, we treated the graph as homogeneous during the denoising inference, considering only the original graph distribution and the noise distribution, thereby enhancing the denoising performance. Regarding the $q(\mathcal{\tilde{G}}|\mathcal{G})$, it is an intermediate variable in the formula optimization. For the overall denoising inference objective, we approximate the objective using the Monte Carlo sampling method. During optimization, a constraint is introduced that forces $q_{\Psi}^{\prime}(\mathbf{Z}|\mathcal{G})$ to approximate the standard Gaussian distribution $p(\mathbf{Z})$.

---

> > ### Author Response · Authors · 2024-11-27
> > **Responses to Reviewer 2RQz**
> >
> > >3. The authors only chose the most basic HGNN architecture which only contains GCN are relational learning modules as the base model. It is unclear why the authors do not adopt more HGNN architectures, especially those meta-path based HGNN, which seems to suffer more from noises. To demonstrate that the proposed explainer is flexible to explain various HGNNs, the authors should conduct experiments on various HGNNs.
> >
> > In the appendix, we have included an introduction to heterogeneous graph neural networks (HGNNs), which will help you better understand the recent developments in this field. The recent heterogeneous graph benchmark, HGB[4], demonstrates that HGNNs can achieve good performance without the need for meta-paths and highlights the limitations of meta-path-based methods. This architecture has been widely adopted in several state-of-the-art heterogeneous graph methods, such as HINormer [5] and PHGT [6]. Therefore, we have drawn upon this architecture and simplified it for our method. The focus of our paper is to highlight the effectiveness of the explainer, rather than improving the performance of HGNNs. Thus, this architecture is sufficient for our purposes. We have demonstrated in the paper that meta-path-based HGNNs can exacerbate the influence of noise, and using a meta-path-based HGNN would have a devastating impact on all baselines, making objective comparisons impossible.
> >
> > >4. The related work section should include heterogeneous graph neural networks and more representative and state-of-the-art GNN explainers.
> >
> > Thank you for your comments. In the related work section, we have added an introduction to state-of-the-art GNN explainers, and in the appendix, we included an overview of the current state of heterogeneous graph neural networks. Relevant works include GOAt [7], MixupExplainer [8], SeHGNN [9], and others.
> >
> > >5. In the experiment, it is unclear how the noises are added. Is it adding in the training set, test set or both?
> >
> > We apologize for any confusion caused by our wording. Noise is added to both the training set and the test set to restore the real scene. We have clarified this in the revised manuscript.

---

> > > ### Author Response · Authors · 2024-11-27
> > > **Responses to Reviewer 2RQz**
> > >
> > > >6. The technical novelty is incremental. The idea of using graph information bottleneck for GNN explainer has been studied and the idea of using variational autoencoder for graph denoising is also explored. The paper just did a simple combination of two existing methods.
> > >
> > > This is the first work to study the impact of noise on heterogeneous graph explainers. We theoretically demonstrate the amplification effect of noise by different heterogeneous graph neural networks. The RHGIB we designed can easily address this issue based on the proven theory. We have constructed a suitable and effective method for this purpose. Furthermore, in contrast to the standard VGAE, we innovatively alleviate the information degradation caused by noise through denoising variational inference, which represents a fundamentally different approach in both functionality and concept. The denoising variational inference model employs a Gaussian mixture model to model the posterior probability $p(\mathbf{Z}|\mathcal{G})$, while the VGAE model uses a Gaussian distribution to model $p(\mathbf{Z}|\mathcal{G})$. Our method is more robust. At the same time, based on theoretical proofs, we followed a neighborhood aggregation-based propagation method to minimize the impact of noise. The type attention-based explanation generator we designed is also innovative, as it integrates heterogeneous semantics into the explanation generation process, enhancing the confidence of the explanations.
> > >
> > > Reference:
> > >
> > > [1] Dai, Enyan, et al. "Towards robust graph neural networks for noisy graphs with sparse labels." Proceedings of the Fifteenth ACM International Conference on Web Search and Data Mining. 2022.
> > >
> > > [2] Jin, Wei, et al. "Graph structure learning for robust graph neural networks." Proceedings of the 26th ACM SIGKDD international conference on knowledge discovery & data mining. 2020.
> > >
> > > [3] Dong, Mingze, and Yuval Kluger. "Towards understanding and reducing graph structural noise for GNNs." International Conference on Machine Learning. PMLR, 2023.
> > >
> > > [4] Lv, Qingsong, et al. "Are we really making much progress? revisiting, benchmarking and refining heterogeneous graph neural networks." Proceedings of the 27th ACM SIGKDD conference on knowledge discovery & data mining. 2021.
> > >
> > > [5] Mao, Q., Liu, Z., Liu, C., & Sun, J. (2023, April). Hinormer: Representation learning on heterogeneous information networks with graph transformer. In Proceedings of the ACM Web Conference 2023 (pp. 599-610).
> > >
> > > [6] Lu, Zhiyuan, et al. "Heterogeneous Graph Transformer with Poly-Tokenization." Proceedings of the Thirty-Third International Joint Conference on Artificial Intelligence, IJCAI-24, International Joint Conferences on Artificial Intelligence Organization. 2024.
> > >
> > > [7] Lu, Shengyao, et al. "GOAt: Explaining graph neural networks via graph output attribution." arXiv preprint arXiv:2401.14578 (2024).
> > >
> > > [8] Zhang, Jiaxing, Dongsheng Luo, and Hua Wei. "Mixupexplainer: Generalizing explanations for graph neural networks with data augmentation." Proceedings of the 29th ACM SIGKDD Conference on Knowledge Discovery and Data Mining. 2023.
> > >
> > > [9] Yang, Xiaocheng, et al. "Simple and efficient heterogeneous graph neural network." Proceedings of the AAAI conference on artificial intelligence. Vol. 37. No. 9. 2023.

---

> > > ### Comment · Reviewer_2RQz · 2024-11-28
> > >
> > > Thanks for your responses. However, the responses still do not address my concerns on the motivation and lacking experiments on more HGNNs.
> > >
> > > 1. For the motivation, the authors seem to have a misunderstanding of my concern. I totally agree that noises would affect the performance of HGNNs, which is not my concern. However, as a post-doc explainer, the explainer should reflect what HGNN uses for classification. If the HGNN uses noisy edges for prediction, the explainer should truly reflect that, which would help model designers understand if the model behaves as expected and if there is any issue in the training data, e.g., noises or biases, for debugging purpose, and help end-users trust and make correct decision. The removal of noisy edges should be done by HGNN itself by developing robust HGNNs or robust self-explainable GNNs. It should not be the aim of a post-hoc explainer.
> > >
> > > 2. As for experiments on more HGNNs, the authors also did not address my concern. I am not asking the authors to improve the performance of HGNNs. Instead, to show the effectiveness of the proposed method and the flexibility of the proposed method in explaining various HGNNs, the authors should conduct more experiments on HGNNs, especially on meta-path-based GNNs, for which the authors claim that suffer from noises most.
> > >
> > > Based on the above concerns, I would like to keep my original rating.

---

### Meta-Review · Area_Chair_epSD · 2024-12-16

**Metareview:**

The reviewers all agree to reject the paper due to its limited novelty, the nuclear motivation, the missing details on the methodology and the insufficient evaluation. The authors have provided detailed rebuttal to the comments of the reviewers. However, the reviewers think their major concerns are not well addressed by the rebuttal.  The reviewer 2RQz still has the concerns on the motivation of the work and the lacking of experiments on more HGNNs. The three other reviewers' concerns remain after reading the rebuttal. Therefore, I recommend to reject the paper.

**Additional Comments On Reviewer Discussion:**

The authors have provided rebuttals, but the reviewers think their concerns are not well addressed and keep their initial scores.

---

### Decision · Program_Chairs · 2025-01-22

Reject